# Wave erosion, frontal bending, and calving at Ross Ice Shelf

Nicolas B. Sartore[1], Till J.W. Wagner[1], Matthew R. Siegfried[2], Nimish Pujara[3], and Lucas K. Zoet[4]

[1]Department of Atmospheric and Oceanic Sciences, University of Wisconsin-Madison, Madison, USA
[2]Department of Geophysics, Colorado School of Mines, Golden, USA
[3]Department of Civil and Environmental Engineering, University of Wisconsin-Madison, Madison, USA
[4]Department of Geoscience, University of Wisconsin-Madison, Madison, USA

**Correspondence:** Nicolas B. Sartore (nsartore@wisc.edu) and Till J.W. Wagner (till.wagner@wisc.edu)

**Abstract.** Ice shelf calving constitutes roughly half of the total mass loss from the Antarctic ice sheet. Although much attention is paid to calving of giant tabular icebergs, these events are relatively rare. Here, we investigate the role of frontal melting and stresses at the ice shelf front in driving bending and calving on the scale $\sim 100$ m, perpendicular to the ice edge. We focus in particular on how buoyant underwater "feet" that protrude beyond the above-water ice cliff may cause tensile stresses at the base of the ice. Indirect and anecdotal observations of such feet at the Ross Ice Shelf front suggest that the resulting bending may be widespread and can trigger calving. We consider satellite observations, together with an elastic beam model and a parameterization of wave erosion to better understand the dynamics at the ice-shelf front. Our results suggest that on average frontal ablation rather consistently accounts for $20 \pm 5$ m/yr of ice loss at Ross Ice Shelf, likely mostly due to wave erosion and smaller-scale, $\mathcal{O}(100$ m$)$, foot-induced calving. Observational evidence suggests that sporadic larger events can skew this rate (we document one foot-induced calving event of size $\sim 1$ km). Stresses from foot-induced bending are likely not sufficient to initiate crevassing but rather act to propagate existing crevasses. In addition, our results support recent findings by Buck (2024) that additional bending moments, likely due to temperature gradients in the ice, play a role in driving frontal deflections. The highly variable environment, irregularity of pre-existing crevasse spacing, and complex rheology of the ice continue to pose challenges in better constraining the drivers behind the observed deformations and resulting calving rates.

## 1 Introduction

High-emission climate model scenarios project that likely mass loss from the Antarctic ice sheet may raise global mean sea level by up to 45 cm by 2100, relative to the 1994–2014 average (Pattyn and Morlighem, 2020; Seroussi et al., 2020; Fox-Kemper et al., 2021). Beyond sea-level rise, the associated meltwater input alters the temperature and stratification of the Southern Ocean with impacts on the global climate (e.g., Golledge et al., 2019; Jeong et al., 2020; Li et al., 2023). These increases in melt represent a sufficiently substantial modification of the Southern Ocean system that they should be included as a historical forcing in climate simulations (Schmidt et al., 2023).

Antarctic ice mass loss occurs mainly through the ablation of ice shelves, which is dominated by two processes: basal melting and calving. For the two largest ice shelves, Ross and Filchner-Ronne, calving is assessed to be responsible for at least 50% of the mass loss, reaching close to 100% for the Western Ross Ice Shelf (Rignot et al., 2013; Greene et al., 2022).

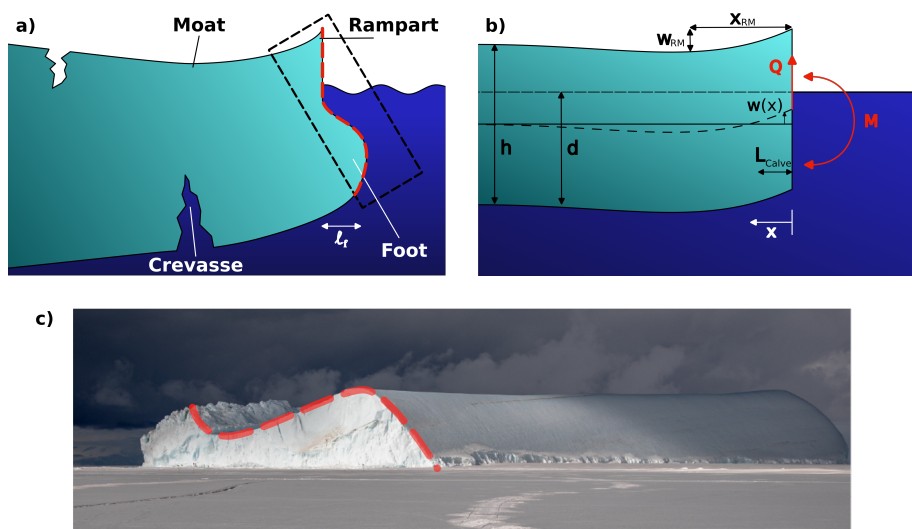

**Figure 1.** Foot-induced deformation of an ice shelf front. (a) Schematic of an ice shelf front deflected by an underwater buoyant foot of cross-edge dimension $l_f$, leading to basal crevasse propagation (not to scale). (b) Elastic beam approximation, where $w$ (dashed curve) is the ice shelf center-line profile relative to the undeflected shelf (solid line), $Q$ is a point force and $M$ a bending moment applied at the front. $L_{\text{calve}}$ is the calving size computed from the point of maximum stress. In reality, $L_{\text{calve}}$ would be determined by a complex interplay between crevassing and the applied stresses. (c) Photo taken in 2019 by Justin Lawrence (used with permission) of an iceberg that calved off western Ross Ice Shelf near 166° E and was subsequently frozen into sea ice. The iceberg likely rotated after calving because of the excess buoyancy of the foot, leading to the smooth and previously submerged part of the ice cliff to be visible. This part exhibits a foot (with $l_f$ several tens of meters) beneath the more rugged above-water ice cliff. The approximate visible part is indicated as a black dashed rectangle in panel a and the corresponding frontal profiles in panels a and c are indicated by the red dashed line. The horizontal along-front width of the iceberg is roughly 500 m.

Ice shelf calving has received substantial attention in recent years, with advances in modeling and observational approaches (see reviews by Benn et al., 2017; Alley et al., 2023; Bassis et al., 2024). Modes of calving range from frequent small-scale failure of the above-water ice cliff to sporadic detachments of giant tabular icebergs. Here, we focus on processes that have received less attention: frontal wave erosion and resulting calving due to bending stresses in the ice. The role of bending was studied for the ice shelf interior in driving crevasse opening by Buck and Lai (2021). However, bending is also encountered

at ice fronts when melting is not uniform with depth, leading to undercutting or overcutting of the ice cliff. The resulting hydrostatic imbalance leads to bending stresses in the ice and flexure that can cause calving events that are larger than the loss due to frontal melt alone. This has been termed the calving-multiplier effect (e.g., Slater et al., 2021).

Depth variations in frontal melt can result from several processes, such as enhanced basal melt forced by a subglacial discharge plume (Jenkins, 2011) or increased near-waterline melt due to advection of warmer surface waters (Slater et al.,

2018). Here, we focus on ocean surface waves as a primary driver of depth-variable erosion. When the cliff of an ice shelf is exposed to open water, waves melt a notch at the calving front, which over time leads to the gravity-driven collapse of the

overhanging ice slab. The submerged front of the ice shelf then protrudes beyond the above-water cliff and is no longer in hydrostatic equilibrium. The excess buoyancy of this protrusion, or foot, will cause the front of the ice shelf to bend upward. This bending results in a characteristic surface expression that Scambos et al. (2005) termed a rampart–moat profile (Figure 1a). The wave-induced erosion steps repeat several times until the tensile stress from buoyancy-induced bending exceeds the strength of the ice, triggering a calving event. This has been referred to as the "footloose" calving mechanism (Wagner et al., 2014).

Observing the underwater section of tidewater glaciers and ice shelves is often hazardous and little direct data was available until recently. However, new technological advances such as the use of uncrewed vehicles have demonstrated that submerged feet, or more generally overcutting, is a widespread phenomenon, particularly in the relatively warmer settings of Alaska and Greenland tidewater glaciers (e.g., Sutherland et al., 2019; Abib et al., 2023). The rampart–moat surface expression of a buoyant foot is more readily observed than the foot itself. For example, James et al. (2014) observed the progression of a rampart–moat profile at Helheim Glacier before and after a calving event. Wagner et al. (2016) argued that this deformation may be explained by a growing submerged foot. Rampart–moat profiles have also been observed for icebergs, e.g., from ICESat data (Scambos et al., 2005) or ship-based lidar (Wagner et al., 2014). The latter study also revealed direct observations of a coinciding foot using multi-beam sonar for underwater imagery that was paired with the above-water lidar. Since in many cases only the rampart–moat surface profile is observed, the presence of a foot tends to be indirectly inferred, and other possible drivers of the surface deformation exist. One alternative driver are internal stresses that result from strong temperature gradients in the ice shelf, a process recently explored by Buck (2024). Part of the motivation of the present study is to explore whether the characteristic bending due to a foot together with estimated wave-induced melt rates is consistent with recent observations of rampart–moat profiles and calving events at Ross Ice Shelf.

While footloose-type calving has been studied at tabular icebergs (e.g., England et al., 2020; Huth et al., 2022) and tidewater glaciers (e.g., Trevers et al., 2019), its potential impact on Antarctic ice shelves has not been investigated in detail. A recent analysis of satellite altimetry data from NASA's Ice, Cloud, and land Elevation Satellite 2 (ICESat-2) mission by Becker et al. (2021) shows that much of the Ross Ice Shelf front exhibits conspicuous rampart–moat profiles, suggesting that foot-induced bending and calving may be commonplace for much of Ross Ice Shelf and potentially other ice shelves.

Here, we first constrain the rate of frontal ablation using satellite observations. We then compare observed elevation profiles of the ice shelf front with solutions of an idealized elastic beam representation. Next, we estimate the foot growth rates using a parameterization of wave erosion at the ice front. We combine these results to validate the beam model and test the wave erosion parameterization, and finally estimate the potential calving frequency and volume at Ross Ice Shelf due to foot-induced flexure.

## 2   Motivating observations of Ross Ice Shelf surface profiles from ICESat and ICESat-2

The underwater section of the Ross Ice Shelf front has not been observed in situ, making it challenging to directly verify the existence or shape of an underwater foot. The photo in Figure 1c of an iceberg that capsized after calving, revealing the distinct

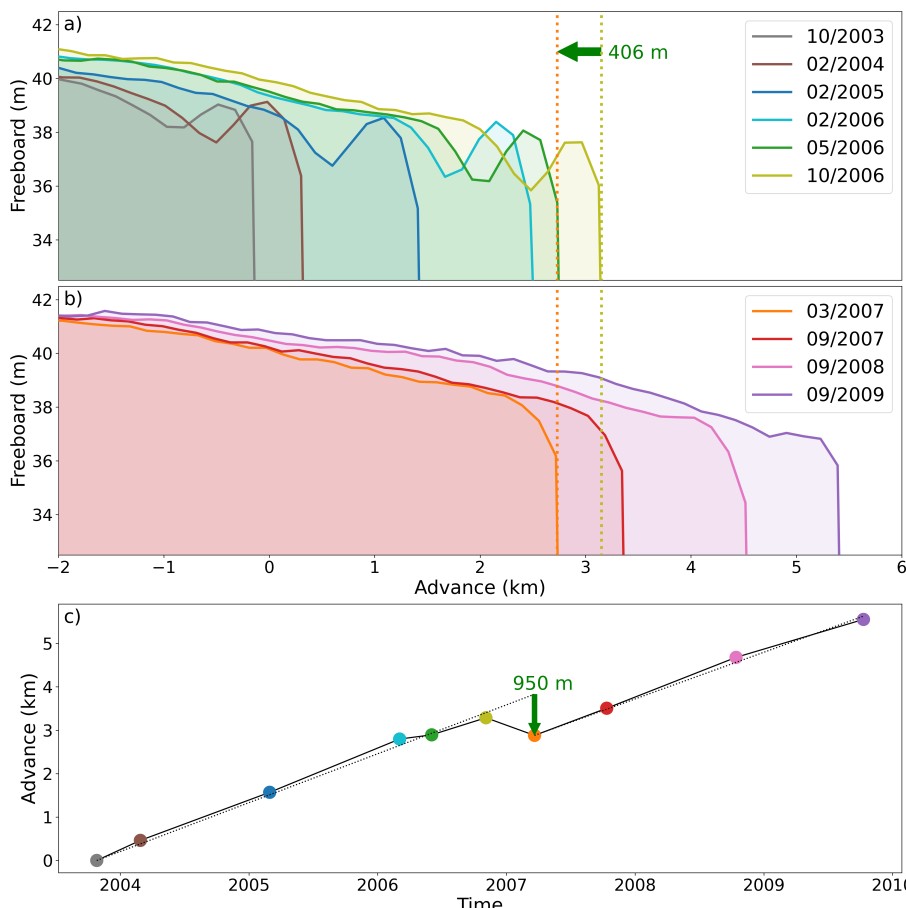

**Figure 2.** ICESat elevation data of the Ross Ice Shelf near-front region at 178.8°E. (a) Six profiles of freeboard elevation within 2–5 km of the ice front, collected between October 2003 (gray) and October 2006 (yellow). The horizontal axis is set to zero at the front of the earliest profile. Clearly visible is the presence and growth of the rampart–moat shape. (b) Four elevation profiles collected between March 2007 and September 2009. The first two feature a standard berm shape, while the rampart–moat structure reemerges in the final two profiles. Between the October 2006 and February 2007 profiles, the front retreated by around 406 m (see vertical lines in panels a and b). This suggests a calving event of roughly 950 m, since the glacier also advanced 544 m in the intervening 4 months, assuming a nearly constant frontal advance speed of 1000 m/yr. This speed is estimated from a linear fit to the frontal advance plot in panel (c).

profile of an underwater foot presents a rare exception and provides perhaps the strongest existing direct evidence of such a foot at Ross Ice Shelf.

Ice-surface profiles from NASA's ICESat (2003–2009) and ICESat-2 (2018–present) laser altimetry missions provide two high-accuracy datasets that yield striking insights:

1. Figure 2 shows 10 repeat ICESat transects (track 0068 of the L2 Global Antarctic and Greenland Ice Sheet Altimetry data
product, GLAH12, release 34; Zwally et al., 2014), with corrections for tides using CATS2008 (Padman et al., 2008;

Howard et al., 2019), inverse barometer effects (Padman et al., 2003), and Gaussian-centroid bias (Borsa et al., 2014). The transects cross the Ross Ice Shelf front at 77.8 °S, 178.8 °E and were collected at roughly equal time intervals over 6 years. This time series appears to capture a rampart–moat growth and calving cycle: starting in late 2003, the rampart–moat structure is clearly visible and becomes steeper over time, until a calving event occurs in late 2006, resetting the frontal profile to a classic berm shape and causing a retreat of the front of $\approx 1$ km (Figure 2c). Following the calving event, the front advances again at the same speed as before the event, and a new rampart–moat starts to form by 2009/10.

2. Using transects of ICESat-2 data collected between October 2018 and July 2020, Becker et al. (2021) showed that the rampart–moat shape is a characteristic feature found along approximately three quarters of the Ross Ice Shelf front. The presence of this smoothly undulating shape suggests that Ross Ice Shelf may have an underwater foot for much of its calving front. We manually classified the 3480 transects of the Becker et al. (2021) dataset according to the extent of near-frontal surface deflection: 2318 transects were excluded, either because they did not cross the front, featured large data gaps, or were not readily classifiable due to large crevasses that resulted in substantial elevation uncertainties. Among the 1162 remaining transects, 220 ($\sim20\%$) were found to feature downward-sloping berm profiles, and 928 ($\sim80\%$) transects exhibited rampart–moat shapes (Figure A1). Here, we will analyze the ICESat-2 transects that exhibit a rampart–moat shape and compare these to an elastic beam model. Analyzing the berm deformations is challenging, in part because the decrease in ice freeboard when approaching the front can be caused by both an decrease in ice thickness and by downward bending at the front (Figure A2). Distinguishing between the two effects is not readily feasible with the methods used here.

For ICESat, the accuracy is 14 cm and the precision 2.1 cm (Shuman et al., 2006). For ICESat-2, accuracy is 3 cm and precision 9 cm (Brunt et al., 2019). Rampart–moat deformations are typically detected on $\sim 1-10$ m vertical scales, which suggests both satellites have sufficient accuracy and precision for the present purpose. ICESat-2 surpasses ICESat in two key aspects: footprint size (12 m vs. 70 m) and spatial resolution (40 m vs. 170 m). ICESat-2 therefore provides a much finer resolution of the rampart–moat profiles (which typically have a horizontal extent of a few hundred meters).

## 3   Methods

### 3.1   Elastic beam representation

To gain physical insight into the deflection and calving process, we consider the idealized representation of the near-front ice shelf as a two-dimensional semi-infinite elastic plate of uniform thickness. Neglecting along-front variations, the model reduces to a 1-D elastic beam equation (Mansfield, 1964). A uniform buoyancy–weight force is applied along the beam, a point force at the front represents the effect of the foot, and a frontal moment is added to model internal and external bending stresses. Implications of the various simplifying assumptions, such as a purely elastic rheology, uniform thickness, and lack of crevassing are discussed below. The hydrostatic balance equation for such a floating beam of uniform thickness $h$ can then be

written as (e.g., Vella and Wettlaufer, 2008; Wagner et al., 2014):

$$B\frac{\mathrm{d}^4 w}{\mathrm{d}x^4} = \rho_w g\left(h/2 - w\right) - \rho_i g h + Q\delta\left(x\right) \tag{1}$$

where $x$ is the distance perpendicular to the front, $w(x)$ the deflection of the beam centerline relative to the unperturbed isostatic equilibrium (see Figure 1b), $\rho_i$ the density of ice, $\rho_w$ the density of water and $g$ the acceleration due to gravity. The flexural rigidity (or bending stiffness) of the beam is defined as $B \equiv \frac{1}{12}Eh^3/(1-\nu^2)$, where $E$ is the elastic modulus. Poisson's ratio $\nu$ is fixed at $\nu = 0.3$, a typical value for ice (Vaughan, 1995); previous studies by Christmann et al. (2016) and Mosbeux et al. (2020) have found that changing to a larger Poisson's ratio of 0.4 or 0.5 tends to have a small effect of $< 5\%$ on the magnitude of maximum tensile surface stress. The first term on the right of (1) gives the upward acting buoyancy force. The second term on the right represents the weight of the beam, and $Q\delta(x)$ describes the foot-induced point force acting at the glacier front ($x = 0$), with $\delta(x)$ the Dirac delta function. We assume an idealized full-depth rectangular foot of cross-sectional dimensions $l_f$ and draft $d = h\rho_i/\rho_w$ (thickness of the submerged ice in isostatic equilibrium), such that $Q = g\left(\rho_w - \rho_i\right)dl_f$.

In order to account for bending stresses at the front we impose a bending moment $M$, giving the boundary condition $B\left.\frac{d^2 w}{dx^2}\right|_0 = M$. Bending stresses may arise through several processes. Most well known is the external downward bending moment that arises from a horizontal imbalance between ice and water pressures at the front (as described by Reeh, 1968). Deviations from a vertical face, due to over- or undercutting can add to to this moment (Slater et al., 2021). Finally, in recent work, Buck (2024) showed that the front may also be experiencing upward bending due to vertical viscosity gradients in the ice (a result of large temperature differences between the cold top surface and relatively warm base of the ice shelf). The resultant net frontal bending moment, $M$ can be upward (positive in our reference frame) or downward (negative), depending on which process dominates. Its exact value is difficult to estimate a priori, and we treat $M$ as a tuning parameter in our model.

We fill refer to the model with both a foot and a frontal moment as the "full model", while the term "foot-only model" describes the limit with only a foot and $M = 0$ (as in, e.g., Wagner et al., 2014).

Applying clamped boundary conditions at $x \to \infty$, Equation (1) can be solved for the near-front deflection. This results in the well-known form of an exponentially decaying horizontal oscillation (e.g., Hetényi and Hetbenyi, 1946):

$$w(x) = e^{-\frac{x}{\sqrt{2}l_w}}\left[\left(l_w\mathcal{M} + \sqrt{2}l_f\mathcal{H}\right)\cos\left(\frac{x}{\sqrt{2}l_w}\right) - l_w\mathcal{M}\sin\left(\frac{x}{\sqrt{2}l_w}\right)\right], \tag{2}$$

where the characteristic buoyancy wavelength is defined as $l_w \equiv \left(B/\rho_w g\right)^{1/4}$, a measure of the energetic balance between beam bending and displacing water. Here, $\mathcal{H} \equiv \left(1 - \rho_i/\rho_w\right)d/l_w$ is a non-dimensional scaling factor related to the vertical dimension of excess buoyancy, such that the product $l_f\mathcal{H}$ determines the magnitude of the upward lift at the front induced by the foot. The non-dimensional moment is defined as $\mathcal{M} \equiv l_w M/B$. We note that the frontal curvature, $\left.\frac{d^2 w}{dx^2}\right|_0 = M/B = \mathcal{M}/l_w$ is independent of the foot length, $l_f$. The sign of the curvature is therefore dictated by the sign of $M$; a negative moment results in a concave front, while a positive moment leads to a convex front. Solution (2) has the same form as in Slater et al. (2021), who considered the opposite role of *undercutting* at glacier fronts (with consistently negative $Q$ and $M$).

For the observed ICESat(-2) surface profiles, we extract the horizontal distance between the ice front and the "moat location", $x_{RM}$, defined at the maximum depression (i.e., at the center of the moat). To do so, each transect is projected onto the meridian

of its mean longitude. Since most of the ice shelf front is close to zonal in its orientation, the meridional projection of a transect ensures that the profile runs approximately perpendicular to the ice front. We also measure the total vertical rampart–moat height difference $w_{RM} = w(0) - w(x_{RM})$, as indicated in Figure 1. These observed quantities can be compared to the beam model, since theoretical expressions for $x_{RM}$ and $w_{RM}$ are obtained from (2). In the full model (with foot and bending moment) the expressions are somewhat cumbersome (not shown). In the foot-only limit (for small moments or long feet) they

reduce to:

$$x_{RM} = 3\frac{\pi}{2\sqrt{2}}l_w, \tag{3}$$

$$w_{RM} = \left(\sqrt{2} + e^{-3\pi/4}\right)l_f\mathcal{H} \approx \sqrt{2}l_f\mathcal{H}. \tag{4}$$

Note that in this limit the location $x_{RM}$ depends on the flexural rigidity alone (through $l_w$) and not on the size of the foot. The frontal uplift $w_{RM}$ on the other hand scales with the foot volume $l_f d$ (per unit lateral width), and inversely with the buoyancy

length $l_w$ (through $\mathcal{H}$).

The stresses induced by bending will be largest at the bottom and top surfaces of the beam and reach a maximum at a distance $L_{\text{calve}} = \pi/(2\sqrt{2})l_w = x_{RM}/3$ from the ice front, which is the locus of maximum curvature (still in the small moment limit). This maximum stress is $\sigma_{\max} = Y \left|\frac{\mathrm{d}^2 w}{\mathrm{d}x^2}\right|_{L_{\text{calve}}}$ , where $Y \equiv \frac{1}{2}Eh/\left(1 - \nu^2\right)$ is the stretching stiffness of the beam (Mansfield, 1964). Following Wagner et al. (2014), we assume that a calving event will be triggered at $x = L_{\text{calve}}$ when the tensile stress

at the base reaches the yield strength, $\sigma_y$, of the beam, i.e. when $\sigma_{\max} = \sigma_y$. The selection of this simple calving criterion was motivated by the analytical nature of this study. We emphasize that this is a highly idealized representation and more fully resolved accounts of failure limits are the subject of much current research, for example using damage and Linear Elastic Fracture Mechanics (LEFM) approaches (e.g., Duddu et al., 2013; Albrecht and Levermann, 2014; Yu et al., 2017; Gao et al., 2023).

Using this stress balance at the point of calving, and computing the curvature at $x = L_{\text{calve}}$ from (2) in the foot-only limit, Wagner et al. (2014) obtain following expression for the critical foot length to induce calving, $l_f^{\max}$:

$$l_f^{\max} = \frac{e^{\pi/4}}{6} \frac{\rho_w}{\rho_i g \left(\rho_w - \rho_i\right)} \frac{h}{l_w} \sigma_y. \tag{5}$$

The calving event triggered when $l_f$ reaches $l_f^{\max}$ will have the above-water length $L_{\text{calve}}$ and the underwater length $L_{\text{calve}} + l_f^{\max}$.

## 3.2 Wave-induced melting

In this framework, the frequency at which the foot-induced stresses trigger calving is determined by the rate of growth of the foot, i.e., $\mathrm{d}l_f/\mathrm{d}t$. This is closely related to the wave erosion of the ice cliff near the waterline, written as the melt rate, $r = \mathrm{d}m/\mathrm{d}t$, with $m$ the melted distance perpendicular to the ice front. We assume that as waves thermally melt a notch into the cliff the overhanging ice is continuously removed by frequent small-scale serac-type failure of the freeboard. If we further assume that the mean ambient melt of the draft is small compared to the wave-induced near-surface erosion (White et al.,

1980), then the underwater foot grows at the same rate as the waves erode the cliff, i.e., $\mathrm{d}l_f/\mathrm{d}t = r$. The validity of the small

ambient melt assumption will depend on the given environmental conditions. It is likely better satisfied in scenarios with strong temperature stratification and where there is sufficient open water near the ice front for substantial wave genesis. The assumption has been found to generally hold up well for icebergs drifting in open waters (e.g., Wagner and Eisenman, 2017), and we assume that the Ross Sea Polynya (discussed below) may allow for similarly high relative rates in wave-induced melt

versus ambient melt. To our knowledge there is no existing parameterization of wave erosion at ice shelf fronts, so we draw on an empirical expression derived from laboratory experiments for floating ice blocks. Different versions of this have been used in the iceberg decay literature since the 1980ies (e.g., White et al., 1980; El-Tahan et al., 1987; Bigg et al., 1997). We adapt the form in Gladstone et al. (2001), which is an expression of the melt rate in terms of sea surface temperature, $T$, near-surface wind speed, $|\boldsymbol{u}|$, and sea ice concentration, $c$:

$$r = \tfrac{1}{2}\left(\alpha_1 + \alpha_2 T\right)\left(\beta_1\sqrt{|\boldsymbol{u}|} + \beta_2\,|\boldsymbol{u}|\right)\left(1 + \cos\left[\pi c^n\right]\right). \tag{6}$$

We use the empirical parameters from Martin and Adcroft (2010), as written in England et al. (2020): $\alpha_1 = 0.67$, $\alpha_2 = 0.33\,°\mathrm{C}^{-1}$, $\beta_1 = 8.7 \times 10^{-6}\,\mathrm{m}^{1/2}\,\mathrm{s}^{-1/2}$, and $\beta_2 = 5.8 \times 10^{-7}$. Gladstone et al. (2001) propose $n = 3$, which has been adopted in subsequent studies. However, we find that $n = 1$ may be more accurate (discussed below). The wind speed term is invoked to represent wave energy, using a relation between the Beaufort Scale and the sea state (Bigg et al., 1997). Note that (6) is a

185 local parameterization of the wave-induced melt rate, not taking into account non-local processes such as swell generated in the open ocean. We emphasize that (6) has not been validated comprehensively against real-world conditions. This presents an opportunity to test how the parameterization performs against well-constrained ice-shelf ablation rates. We calculate a wave-induced melt rate climatology at Ross Ice Shelf from observed monthly environmental fields $T$, $|\boldsymbol{u}|$, and $c$, provided by the data sets discussed below in Section 3.3. To minimize variability at the ice–ocean boundary and simplify the melt rate to a function

of longitude, we calculate the mean over an ocean strip extending 60 km seaward from the Ross Ice Shelf front (see Figure 3). The resulting melt rate estimates are not overly sensitive to the specific choice of strip width.

### 3.3 Ross Sea environmental data

The melt parameterization (6) incorporates sea surface temperature (SST), near-surface wind speeds, and sea ice concentration (SIC). Here, we use the Group for High Resolution Sea Surface Temperature (GHRSST) product at 0.01° (0.23 km) resolution

(NASA/JPL, 2015) for SST, the ERA-5 monthly reanalysis product (Hersbach et al., 2023) for 10 m surface wind speed (with native horizontal resolution 0.25°/5.8 km), and the National Snow and Ice Data Center Climate Data Record v.4 satellite SIC monthly dataset at 25 km resolution (Meier et al., 2021). All datasets are monthly averaged over the years 2003–2022 to compute a climatological mean estimate of melt rates, and they were regridded to the regular GHRSST 0.01° grid without interpolation.

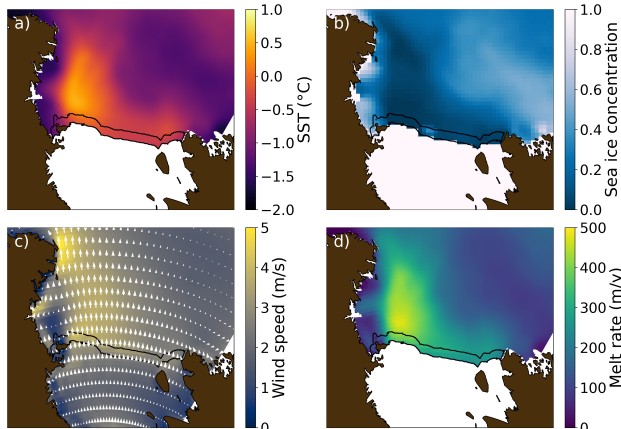

**Figure 3.** Environmental properties in the Ross Sea and corresponding local melt rate estimate derived from the listed observations and reanalysis. Shown are January fields averaged over 2003–2022 for (a) SST (GHRSST from NASA/JPL, 2015), (b) SIC (NSIDC Climate Data Record from Meier et al., 2021), (c) wind velocities (ERA-5 from Hersbach et al., 2023), and (d) melt rate computed from the other fields using Equation (6). The 60 km near-frontal strip over which the environmental variables are averaged is indicated in all panels (black contour).

## 3.4  Total ablation estimate

The rate of frontal ablation, $A$, represents the cumulative effect of frontal melting and calving events that result in the removal of ice from the front of ice shelves. This quantity can be estimated from observations as the difference between frontal advance velocity, $F$, and ice flow velocity near the front, $V$, such that $A = V - F$.

**To obtain near-front ice flow velocity ($V$):** Klein et al. (2020) deployed 12 GPS stations from November 2015 to December 2016, spanning from the front of the Ross Ice Shelf to 430 km upstream. Three of these stations were located $\sim$1 km from the ice shelf front: DR01 at (178.35° E, 77.77° S), DR02 at (178.43° W, 77.82° S), and DR03 at (175.12° W, 78.26° S), providing three high-accuracy estimates of near-front velocities.

**To obtain frontal advance velocity ($F$):** The Sentinel-1 C-band Synthetic Aperture Radar satellite (Copernicus, 2015, 2022) provides imagery of the Ross Ice Shelf front from 2015 to the present on a sub-monthly timescale. After geolocating the data, we manually extracted the front position for all available images that overlap with the locations of the three buoys (DR01-03). The extracted positions were chosen as the intersections of the ice front with the direction vector of the buoy velocity. This process resulted in three time series of the frontal position from 2015 to 2024 along the buoy flow lines with a minimum of 126 data points per series.

## 3.5  Calving frequency and volume

Some Antarctic ice shelves are marked by regularly spaced crevasses (see, e.g., front of the Thwaites Glacier, Figure A3). In these cases, calving rates are understood to be determined by crevasse spacing, ice flow speed, and associated ice shelf

thinning, which will eventually lead to tensile stresses that are large enough to open up the crevasses such that calving occurs (e.g., Buck, 2023). For steady ice velocities this would suggest regular calving events of a given characteristic size (set by the crevasse spacing). For a given calving frequency $f$ and characteristic calving length $L_{\text{calve}}$ in the direction of flow, the rate of ice loss from calving, $C$ (retaining the assumptions of uniform thickness and no along-front variability) is then simply $C = f\,L_{\text{calve}}$. Here, $C$ is measured as distance per unit time of ice lost in the direction perpendicular to the ice front.

For Ross Ice Shelf, large crevasses are much rarer and unevenly spaced (Figure A3). While the frontal ice loss is likely dominated by infrequent calving of giant icebergs, it is unknown to what degree smaller-scale calving events play a role (sometimes referred to as edge-wasting; Scambos et al., 2005), which we suggest include footloose-type calving. For footloose calving, we combine the beam model with the estimated melt rates from (6), assuming that calving occurs each time when the melt distance $m$ is equal to a foot of size $l_f = l_f^{\max}$. This gives a calving frequency $f = r/l_f^{\max}$. The time-averaged ice loss rate due to footloose-type calving is then written as

$$C = \frac{r}{l_f^{\max}} L_{\text{calve}}. \tag{7}$$

This allows an assessment of how the footloose-induced calving rate depends on environmental factors and ice thickness, as well as on the material properties $B$ and $\sigma_y$. It further enables us to put this calving process in relation with the total frontal mass balance of the ice shelf.

## 4  Results and Discussion

### 4.1  Observations of frontal ablation

Klein et al. (2020) showed that for the three buoys DR01-03, both intra-annual and inter-annual variations were small compared to the mean velocity, resulting in steady velocities: $V = 1029 \pm 1$ m/yr for DR01, $1100 \pm 6$ m/yr for DR02, $1018 \pm 3$ m/yr for DR03. Similarly, the front positions derived from Sentinel-1 show a an approximately steady advance of the Ross Ice Shelf front from 2015 to 2024, with both intra-annual and inter-annual variability being small compared to the mean frontal advance velocity. The observed frontal advance is shown in Figure A4, with $F^{\text{obs}} = 1012, 1074, 997$ m/yr for DR01-03, respectively. The absence of large-scale calving signals suggests that such events are relatively rare and that total ablation primarily consists of continuous melting and small-scale calving ($\sim$ 100 m or less). Figure 4 compares the front advance velocity fitted from Sentinel-1 data (blue) to the ice flow velocity from the buoys (red). The difference $(V - F)$ gives the total ablation rate at each location: $A = 16.7$ m/yr for DR01, $25.5$ m/yr for DR02, $20.1$ m/yr for DR03. This entails that roughly 1.5–2.5% of the ice transported to the shelf front is lost through continuous melt and small-scale calving.

We note that near-front ice velocities obtained from MEaSUREs Version 2 (Rignot et al., 2017) tend to be lower than those measured by the buoys (Figure 4). This is particularly evident for DR01 and DR03, where unphysically low MEaSUREs estimates are found, as the ice flow velocity cannot be slower than the frontal advance velocity. This discrepancy is presumably due to the relatively low resolution of MEaSUREs (450 m). Finally, in brown, we present the estimated frontal advance, $F^{\text{est}}$ computed by subtracting the estimated melt (Equation 6) from the buoy velocity. This gives $F^{\text{est}} = 952, 1039,$ and $948$ m/yr

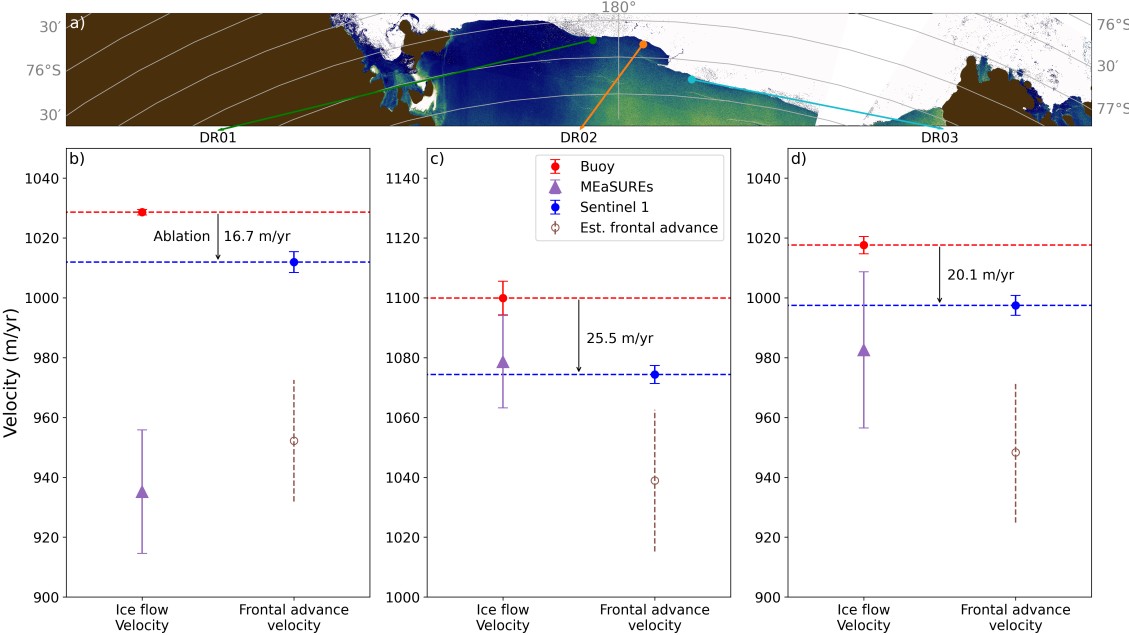

**Figure 4.** (a) Map of the Ross Ice Shelf near-front region. Indicated are the locations of GPS buoys DR01–03 from Klein et al. (2020) in green, orange, and cyan, respectively. (b-d) Comparison of annual mean *ice flow velocity* (left) and *frontal advance velocity* (right) at the locations of (b) DR01, (c) DR02, and (d) DR03. Shown is the ice velocity from the GPS buoy, located within $\sim 1$ km from the ice front (red) and the frontal advance velocity at that location as extracted from Sentinel-1 imagery (blue). The red error bars show inter-annual variability in the buoy data, the blue error bars show uncertainty in the frontal advance velocity, estimated using a parametric bootstrap method. The difference between the ice velocity and the frontal advance velocity gives an estimate of annual mean frontal ablation (black arrow). Also shown are frontal ice velocity estimates from MEaSUREs (purple). In brown we show the estimated frontal advance, $F^{\text{est}}$, computed by subtracting the estimated melt (Equation 6) from the buoy velocity. The brown vertical dashed lines illustrate the inter-annual variability in the melt estimate.

for buoys DR01–03. The significant discrepancy between the estimated advance $F^{\text{est}}$ (brown) and the observed advance $F^{\text{obs}}$
(blue) implies that Equation 6 overestimates the melt rate (discussed below).

## 4.2   Beam theory fit to observations

We first assess whether the idealized floating beam subject to a point force and bending moment at its front describes an ice shelf that is consistent with the satellite observations. We do so by comparing the beam solution (2) to the ICESat-2 transects that were identified as featuring rampart–moat profiles.

In order to compare ice shelf segments with different rampart–moat heights and horizontal extents, we align and normalize the observed profiles. We first shift all profiles vertically so that the moat location is at $w(x_{RM}) = 0$. The horizontal dimension is then scaled by an observational estimate of $l_w^{\text{obs}} = 2\sqrt{2}/(3\pi)x_{RM}^{\text{obs}}$, obtained from the foot-only limit (3). The vertical

dimension is scaled by $w_{RM}^{\mathrm{obs}}$, which is given by the vertical difference between the ice front $w(0)$ and the central moat depression $w(x_{RM})$. In Figure 5 we show the resulting dimensionless ICESat-2 profiles, together with the dimensionless solution $W(X)$ of Equation (2) with no bending moment ($M = 0$). Here, $W = w/w_{RM}$ and $X = x/l_w$. The moat location for both the theoretical and normalized profiles is at $X_{RM} = 3\pi/(2\sqrt{2})$.

Figure 5a shows general agreement between the transects and the beam solution. Figure 5b and 5c show that the moat position ranges between $x_{RM} = 50 - 750$ m (with most values $100 - 500$ m) and the frontal uplift is $w_{RM} = 2 - 15$ m. For this figure we excluded transects that feature downward-sloping berm profiles (since there are no scaling factors $x_{RM}^{\mathrm{obs}}$ and $w_{RM}^{\mathrm{obs}}$ for berms). Berm profiles represent 20% of the data, and as shown in Figure A1, berm profiles (or small ramparts) are typically observed in patches along the Ross Ice Shelf front. This pattern may suggest that local factors, such as high basal melt, prevent the formation of the foot, or that recent calving events have locally removed any trace of it. An example of a berm profile can be seen in Appendix A2.

It is apparent from Figure 5 that the ice shelf thickens with distance from the front, indicated by the increasing surface elevations with increasing $x$. The theoretical solution for a fixed thickness beam with $w \to 0$ for $x \to \infty$ and the observed surface profiles therefore diverge as $x$ becomes large, leading to a vertical mismatch of up to 6 m at a distance of 1500 m from the ice front. Vertical scales in Figure 5 are greatly amplified, and under the assumption of isostatic balance the results above suggest that the ice shelf thins by less than 40 m/km = 0.04 near the ice front (but away from the rampart–moat). The assumption of uniform thickness should thus be largely satisfied near the ice front.

We also provide in Figure 5 two examples of theoretical normalized curves with nonzero negative and positive bending moments $\mathcal{M}$ (black dashed). Incorporating this moment into the model enables us to capture the entire variety of profiles by matching the curvature in the rampart sections of the profiles. Figure 6 shows four individual ICESat-2 profiles, exhibiting a range of uplift and frontal curvature features. The profiles are fitted using the full model (in green) and the foot-only model (in red). The full model is fitted using all data points near the front to estimate $M$, $l_f$, and $l_w$. The foot-only model derives $l_w$ and $l_f$ from the measurements of the uplift ($w_{RM}$) and the moat position ($x_{RM}$) using Equations (3) and (4). Overall, the full model captures the frontal deformation more accurately, in particular for small deflections. Figure 6a shows a profile with a small uplift ($w_{RM}$ = 0.6 m) and a maximum uplift located away from the front. The foot-only model (in red) is not able to capture the slope inversion. By allowing for a negative frontal moment in the full model this feature is reproduced, resulting in a close fit for the full frontal region. This example shows that the combination of a foot and a negative moment is required to explain this type of profile, as a positive moment or foot alone can not reproduce the frontal slope inversion.

Figure 6b illustrates a situation with a slightly larger uplift ($w_{RM}$ = 0.9 m) and a concave frontal shape; again, this is best matched with a non-zero foot and a small negative moment. Figures 6c and 6d depict situations that appear identical, with both cases showing a large uplift ($w_{RM} \approx 10$ m) and a good fit from the foot-only model (red), with only a slightly better fit from the full model. However, the two situations are different: in Figure 6c, the frontal curvature is negative, resulting from a negative moment, with the uplift being a consequence of a large foot (30 m). In Figure 6d, the frontal curvature is positive, and the uplift is entirely determined by moment deformation with no foot.

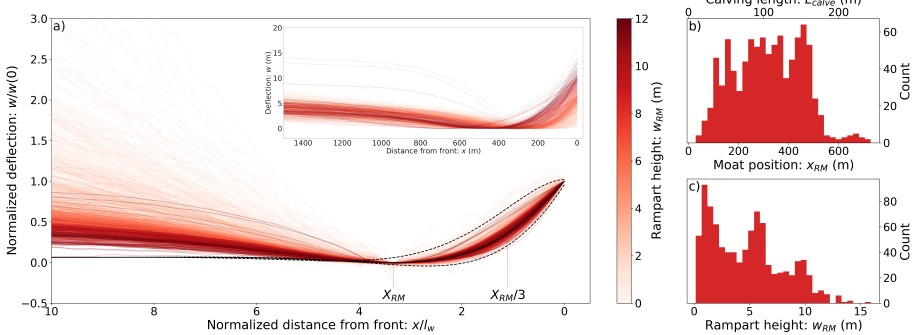

**Figure 5.** Ross Ice Shelf front elevation profiles. (a) Normalized ICESat-2 transects (thin colored lines), with ice front at $x = 0$, vertically shifted such that $w(x_{RM}) = 0$, with dimensionless 1D elastic beam solution in black. The divergence in lighter curves with distance from the ice edge is a result of these transects being vertically scaled by smaller values of $w_{RM}$. The solid black curve shows the foot-only model and the dashed lines show results with additional frontal bending: the upper curve has a negative bending moment $\mathcal{M} = -0.1$ and the lower curve has positive $\mathcal{M} = 0.2$ (for both examples the specified foot-length is $l_f/l_w = 1$; the normalized foot-only model is independent of $l_f$). Also indicated by dotted vertical lines are the location of maximum depression, $X_{RM}$, and the location of maximum stress, $X_{RM}/3$. Inset: same as main figure, but without normalization. (b) Histogram of moat positions ($x_{RM}$) and (c) histogram of rampart heights ($w_{RM}$), both corresponding to transects in panel a.

However, when comparing the two fits of Figure 6d —with and without the additional moment— distinguishing between a large foot and a large positive moment is challenging for large ramparts. This difficulty arises because the curvature induced by a moment is barely noticeable in the presence of a large rampart. Consequently, for around 200 transects (out of the total 928), the surface elevation profiles do not allow us to conclusively determine the relative importance of a foot versus a bending moment in the observed upward deflection.

The results in Figures 5 and 6 are subject to the assumption that the buoyancy length $l_w^{\text{obs}}$ can be treated as a free parameter that is independently fitted for all individual transects. The resulting distributions of $l_w$ are shown in Figure 7a. Both models show similar buoyancy wavelengths, with a mean value of $l_w \approx 100$ m and standard deviation of $\pm 38$ m. Beam theory states, however, that the buoyancy length is determined by the elastic modulus, $E$, and ice thickness, $h$, such that $l_w \propto E^{1/4}h^{3/4}$ (see above). Since $E$ is typically considered a known material parameter and since $h$ is inferred from the observed freeboard, it may be expected that $l_w$ can be constrained independently. To test this, we consider an elastic modulus value used in the literature for ice shelves, $E = 1$ GPa (e.g., Vaughan, 1995; Banwell et al., 2019), which is about an order of magnitude lower than laboratory values for pure ice. We note that values of $E \sim 1$ GPa are typically inferred from tidal flexure near the grounding line, and the effective modulus near the calving front may be different. We estimate $h$ at the front from the observed freeboard for each transect using a depth-averaged ice density of 850 kg/m$^3$, taking into account the less dense firn layer (Drews et al., 2016). Computing $l_w$ this way, we find values of $530 \pm 90$ m, larger than the fitted $l_w$ by a factor of 4 to 9.

Similar discrepancies have been encountered in previous studies that apply an elastic framework to frontal ice shelf bending (Scambos et al., 2005; Wagner et al., 2016; Mosbeux et al., 2020). We suggest that this is due to two main factors: (1) the ice

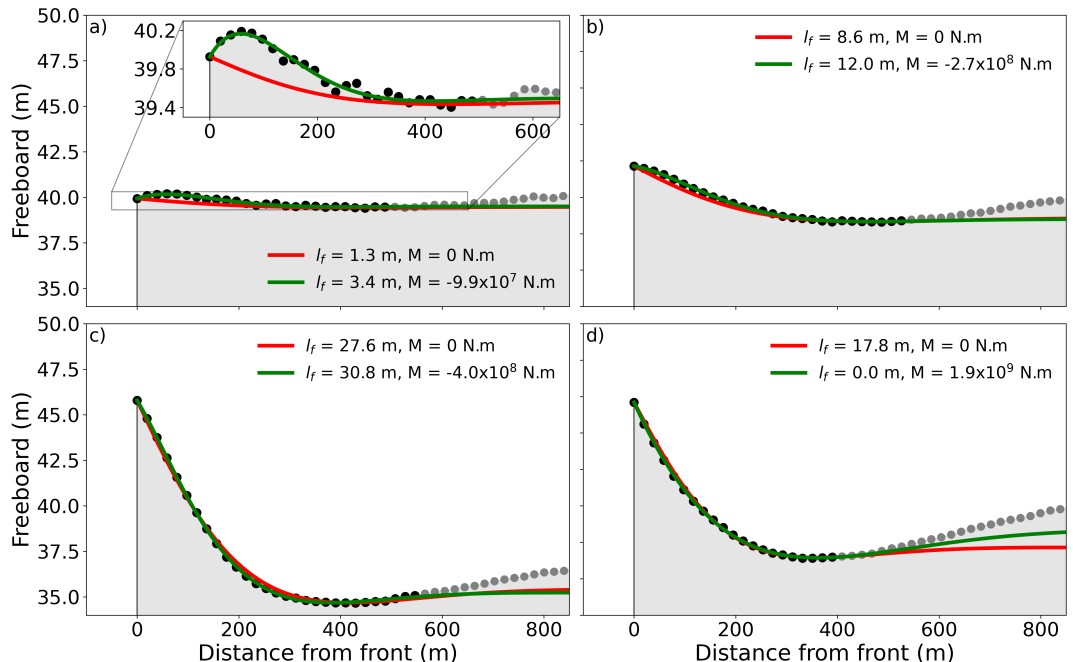

**Figure 6.** Four ICESat-2 profiles with varying frontal deflections (black/gray markers). In each panel the observed profile is compared to the foot-only model (red) and the full model (green). The full model was fitted only to a subset of the observational data (black markers) to avoid issues from the increasing ice thickness with distance from the front. The foot-only model was fitted to $x_{RM}^{\mathrm{obs}}$ and $w_{RM}^{\mathrm{obs}}$. In panel (a) the inset shows a zoomed-in version of the frontal deflection, highlighting a combination of downward curvature and uplift that is most readily explained by a combination of a negative bending moment and positive shear force. In each panel, the figure legend specifies the $l_f$ and $M$ parameters chosen to produce the best fit with the data. By design $M = 0$ for the simple-model (red). Panel (d) highlights the case where two explanations give a close fit: one with a sizeable foot and no bending moment, and one with no foot but a positive bending moment. We argue that disentangling these two processes may act as motivation for further investigations. The transect locations are (from a–d): (-78.25° N, -174.70° W), (-78.31° N, -171.79° W), (-77.39° N, 172.62° W), (-78.00° N, -160.11° W).

undergoes viscous creep on the timescale of rampart–moat development (i.e., years; Figure 2) and plastic failure, impacting the deformation on the time scales relevant here, as discussed further below; (2) the ice shelf is not a uniform and homogeneous beam, but rather features crevasses, smaller-scale damage, a firn layer, temperature gradients, and more. These factors predominantly act to reduce the flexural rigidity, $B$, and thereby the buoyancy length of the ice shelf, relative to that of a perfect beam of ice (Mosbeux et al., 2020). This has been used to suggest an "effective" elastic modulus (Cuffey and Paterson, 2010) or

an effective ice thickness (Scambos et al., 2005) that may be substantially lower than standard values. Mosbeux et al. (2020) propose an effective elastic modulus $E^*$ as low as 2 MPa for the Ross Ice Shelf front, which leads to a reduction in $l_w$ by roughly 80% (compared to $E = 1$ GPa) and brings the theoretical values of $l_w$ in line with those of Figure 7a. For the following analysis, we will proceed with the observational estimates $l_w^{\mathrm{obs}}$.

## 4.3 Estimation of foot-induced calving length, $L_{\text{calve}}$

The idealized calving condition considered here states that the calving length, $L_{\text{calve}}$, is determined by the location where the maximum stress reaches the ice strength, such that $L_{\text{calve}} = x(\sigma_{\max} = \sigma_y)$. In the foot-only model, the location of maximum stress is independent of the foot length, giving $L_{\text{calve}} = x_{RM}/3$ (see above). In the full model, the location of maximum stress depends on $l_f$. For small feet, $x(\sigma_{\max})$ is larger in the full model than in the foot-only model (by up to a factor of 2), but this shifts to the front as $l_f$ grows. (For small feet, the full model also has a stress maximum at $x = 0$ due to the applied bending moment. However, this stress maximum does not trigger calving). In the limit of large $l_f$, the location of maximum stress from the full model converges to that of the foot-only model. Since foot-induced calving typically requires feet to grow large (Wagner et al., 2014), we use this limit to estimate the calving length from the observed profiles, giving $L_{\text{calve}} = 113 \pm 43$ m (see top horizontal axis of Figure 5b). Here, we have excluded profiles that were identified as purely moment-driven, since their maximum stresses are at $x = 0$.

## 4.4 Estimation of frequency of foot-induced calving events

To establish a rough estimate of a typical calving frequency $f = r/l_f^{\max}$, we first find likely bounds on the critical foot length that triggers calving, $l_f^{\max}$, and then consider the melt rate $r$.

### 4.4.1 Maximum foot length, $l_f^{\max}$

Using the foot-only relation (4) and the observed frontal uplift $w_{RM}$ for each profile, we obtain estimated foot lengths $l_f = 0$–40 m (Figure 7b). This is broadly in agreement with underwater feet observed in other settings (e.g., Wagner et al., 2014), and the upper bound of $l_f^{\max} \approx 40$ m appears consistent with the image of the calved iceberg in Figure 1c. Figure 7b shows that the distribution of foot lengths in the full model is similar to the foot-only model except for one notable difference: the full model features $\sim 200$ profiles for which the deformation is purely due to bending moments (i.e., $l_f = 0$), while the foot-only model identifies only $\sim 70$ profiles with negligible feet ($l_f < 1$ m). However, this discrepancy is expected to have little bearing on calving, which occurs in the large-foot limit.

From the observed profile curvatures we next estimate the distribution of maximum tensile stresses, $\sigma_{\max}$ (Figure 7c). Again we find good agreement between both models, with most values in the range $\sigma_{\max} = 0$–100 kPa and approximate the ice shelve tensile strength to be $\sigma_y = 80 \pm 20$ kPa. Comparing Figures 7b and c, we expect that this yield strength is typically attained for feet with $l_f = 30$–40 m.

Indeed, using this value for $\sigma_y$ in Equation (5), together with the previously estimated typical buoyancy wavelength $l_w$ and ice thickness $h = 214 \pm 45$ m, we find the maximum foot length to be around $l_f^{\max} = 43 \pm 22$ m, in good correspondence to the estimates above. This number is only weakly sensitive to changes in $h$ and $E$ due to the 1/4 power scaling in (5), but scales linearly with $\sigma_y$. We will use this value of $l_f^{\max}$ for the calculation of a footloose calving frequency, as discussed next. In reality the actual size of calving will likely be determined by pre-existing basal crevasses near the ice front, which the bending stresses

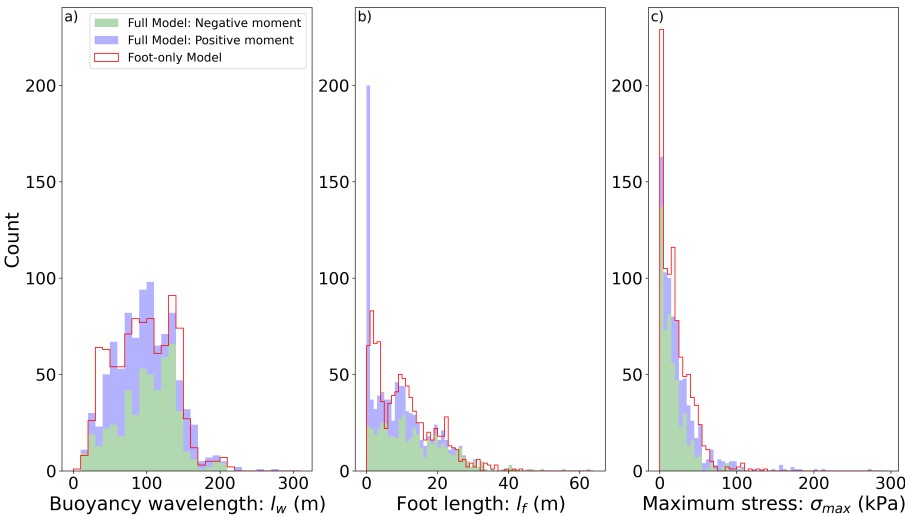

**Figure 7.** Distribution of theoretical buoyancy wavelength ($l_w$), foot length ($l_f$) and maximum tensile stress ($\sigma_{\max}$) for the two model formulations. The full model is depicted in green for negative moments and in blue for positive moments, with the two quantities stacked. The foot-only model is represented in red.

will open up, eventually leading to calving. Requiring relatively low values of $\sigma_y$ to trigger calving in this framework suggests that the bending stresses alone may not be sufficient to initiate crevassing.

### 4.4.2    Wave erosion and foot growth rate, $r$, and calving frequency, $f$

The climatological January fields of SST and SIC in Figures 3a and b show the presence of the large Ross Sea Polynya, extending along most of the Ross Ice Shelf front. This is consistent with the katabatic winds of Figure 3c, blowing down the

ice shelf roughly in parallel with the ice flow direction and pushing the sea ice northward. The polynya allows for greatly enhanced solar heat uptake by the near-shelf ocean. This has been shown to have profound impacts on basal melt rates of the ice shelf (Stewart et al., 2019), but the potential impact on frontal melt has not been studied in detail. The polynya is likely a key factor for wave-induced melting since it allows for both surface heating and for increased wave energy near the front. As a result, Equation (6) estimates January melt rates close to zero to the west of Ross Island where there is substantial sea ice

cover and above 200 m/yr for the rest of the ice shelf where SIC is near zero at this time of year (Figures 3d and 8).

Figure 8 shows the monthly climatological melt rate along the ice shelf front, averaged over the years 2004–2022 and using the 60 km near-front swath indicated in Figure 3d (the along-front monthly-mean melt rates, as functions of longitude, are shown in Figure A5). As expected, melt rates are highest in January (with the along-front mean topping out at $\approx 260$ m/yr) and consistently low in winter, which is due to low $T$ and due to melt rates reducing rapidly because of the $\cos(\pi c)$

term. The along-front mean winter melt is near zero from April through October. The typically used sea ice dependence, $r \sim \cos(\pi c^3)$ from Gladstone et al. (2001) appears to overestimate melt rates at intermediate concentrations ($c \approx 0.5$–$0.8$),

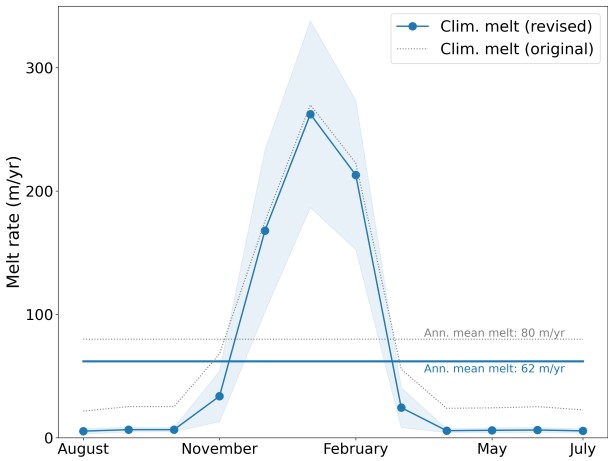

**Figure 8.** Climatological wave-induced melt rate, $r$ at Ross Ice Shelf (2004–2022 mean), computed from the zonal average of along-front melt shown in Figure 3d. The annual mean melt rate is indicated by the horizontal lines. The light blue shaded area shows the monthly standard deviation. Shown are the modified parameterization with $r \sim \cos(\pi c)$ (blue) and the original parameterization $r \sim \cos(\pi c^3)$ (gray dotted).

leading to unrealistically high winter melt rates (around 20 m/yr). Observations of wave attenuation as a function of sea ice concentration (Nose et al., 2020) appear to be better matched by the linear scaling $r \sim \cos(\pi c)$, which we propose as a more faithful parameterization.

The annual mean melt rate from Equation (6) is $r = 62$ m/yr (Figure 8). This is around 3 times greater than the total ablation, $A$, as derived from observations in Section 4.1 (this discrepancy is illustrated in Figure 4). The uncertainty in the observed $A$ is much smaller than that of the melt rate parameterization, which leads us to question the validity of the (by-and-large) untested melt rate (6). Motivated by these findings, we suggest that Equation (6) overestimates $r$ by an order of magnitude, and we find that $r^* = r/10 \approx 6$ m produces more consistent results (see below).

Comparing the rate of foot growth, $r^*$, to $l_f^{\max} \approx 40$ m implies a calving frequency of $f \approx 0.15$ /yr which corresponds to one foot-induced calving event every 6–7 years.

### 4.5    Estimation of characteristic calving rate

Combining the predicted typical calving length, $L_{\text{calve}} \approx 110$ m, with the frequency $f = 0.15$ /yr, we estimate that the annual-mean ice loss due to footloose-type calving at the Ross Ice Shelf is approximately $C = fL \sim 16$ m/yr. Adding this to the frontal
melt rate of 6 m/yr suggests total wave-induced frontal ablation of around 22 m/yr, in agreement with the observations in Figure 4 which showed $A = 20 \pm 5$ m/yr. This is not an independent derivation of the ablation rate, but rather we scaled $r$ by a factor of 10 to produce an estimate of melting plus calving that would be consistent with the observations. This clearly illustrates the need for further work to better constrain the wave-induced melt parametrization. We furthermore emphasize the substantial

spatial and temporal variability in this system, and so these numbers are intended as rough estimates of wave melting and footloose calving for Ross Ice Shelf.

The time series of Figure 2 exhibits some similarities but also differences to these theoretical results: (i) ICESat profiles feature one clear calving event over the 6 year period from 2004–2010, and the observed rampart–moat feature in this case grew over multiple years, which is consistent with the melt rate and calving frequency estimated above. (ii) The observed calving event in late 2006 led to a frontal retreat of about 940 m, larger than the estimated characteristic calving lengths.

We conclude that this calving example is probably not purely foot-driven and that other factors played a significant role in determining the size of the event. A major reason for the discrepancies between observations and theoretical estimates is likely the assumption of purely elastic deformation. Viscous flow almost certainly plays a role on the relatively long timescales over which the foot grows and the rampart–moat profile develops. Mosbeux et al. (2020) provide a detailed study of how viscous versus elastic processes influence the footloose calving mechanism. The authors find that accounting for viscous relaxation will lead to critical stresses being reached more gradually, relative to the elastic framework, and critical foot lengths for a given yield stress are $20 - 30\%$ larger in the viscous framework than the elastic framework. This may explain some of the timescale discrepancies between theory and observations. Notably, Mosbeux et al. (2020) argue that viscous effects lead to the location of global maximum tensile stress moving closer to the ice front as the foot grows, which in turn would cause smaller-size calving events than the elastic case. In this respect, accounting for viscous relaxation would act to reconcile the theoretical estimates with the observed $x_{RM}$ from ICESat-2 data, but not with the large-scale event from the ICESat time series. Other processes, such as the internal bending moments due to thermal gradients mentioned above may constitute important additional controls on the calving cycle.

## 5    Conclusions

The environmental conditions at the front of Ross Ice Shelf are conducive to the development of buoyant underwater feet, and anecdotal evidence such as the image of a calved iceberg near Ross Island (Figure 1c) and the ICESat timeseries in Figure 2 suggest that footloose-type calving may be an important process in controlling the Ross Ice Shelf frontal mass balance. We show that the widespread rampart–moat profiles found in ICESat-2 elevation data can be captured with an elastic beam model that accounts for (i) frontal uplift due a submerged foot and (ii) a bending moment applied at the ice front. While a majority of rampart–moat features are reproduced with a simple foot-only scenario, the model highlights that a subset of profiles are only physically plausible if the foot and bending moment act in conjunction.

Leveraging satellite imagery and GPS buoys we constrain the total ablation at the Ross Ice Shelf front to $20 \pm 5$ m/yr. This is compared to our model results which suggest a characteristic calving size of $L = 113 \pm 43$ m associated with footloose-type calving. We estimate that averaged over time, this process may contribute a loss of $\sim 16$ m/yr along the front of Ross Ice Shelf, in addition to $\sim 6$ m/yr of wave erosion. We further argue that a often-used parameterization of wave erosion likely overestimates the melt rate at Ross Ice Shelf by an order of magnitude.

Compared to a frontal advance of $\sim 1000$ m/yr for much of the central Ross Ice Shelf, our results suggests that frontal melt and edge-wasting may contribute only around 2% of the total mass loss, and that most of the mass balance is controlled by infrequent calvings of giant tabular icebergs. However, under continued future warming and associated increases in sea ice free periods, near-frontal wave energy and ocean heat uptake are expected to increase. This would result in enhanced wave erosion

and small-scale calving rates at the ice shelf front, bringing them closer to the high frontal ablation rates observed at tidewater glaciers in Greenland. Further investigations are therefore warranted to reduce the substantial uncertainties persisting in the estimation of the current and future frontal mass balance of Antarctica's ice shelves.

*Code and data availability.* Code to download data, perform data analysis, plot figures is available at https://github.com/nicsar2/FootlooseCalvingMechanism.git . Sea ice concentration data are available at

425 https://nsidc.org/data/g02202/versions/4 . Wind speed data are available at https://doi.org/10.24381/cds.6860a573 . Sea surface temperature data are available at https://podaac.jpl.nasa.gov/dataset/MUR-JPL-L4-GLOB-v4.1 . Becker et al. (2021) code repository to pre-process ICESat-2 transects is available at https://zenodo.org/records/4697517 . ICESat-2 version 3 ATL06 data are available at https://nsidc.org/data/atl06/versions/6 . GLAH12 release 34 data are available at https://nsidc.org/data/glah12/versions/34 .

## Appendix A: Additional figures

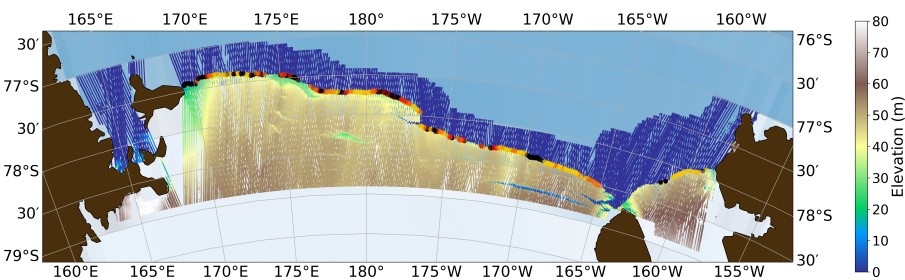

**Figure A1.** Elevation map of the Ross Ice Shelf front from ICESat-2 transects. Rampart–moat profiles detected are marked with a dot, color-coded according to the height: no rampart–moat or below 1 m (black), $1-2$ m (yellow), $2-5$ m (orange), and above 5 m (red). Land is shown in brown.

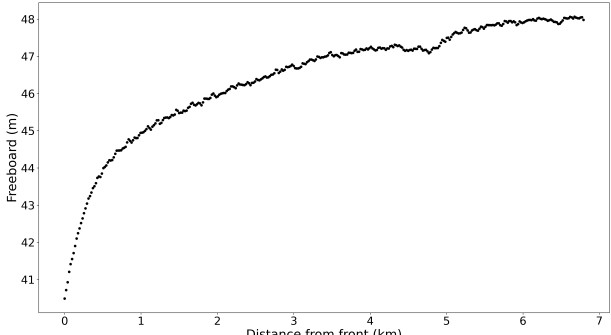

**Figure A2.** Example of typical berm profile from ICESat-2 altimetry located at 78.35 °S, 169.4 °E.

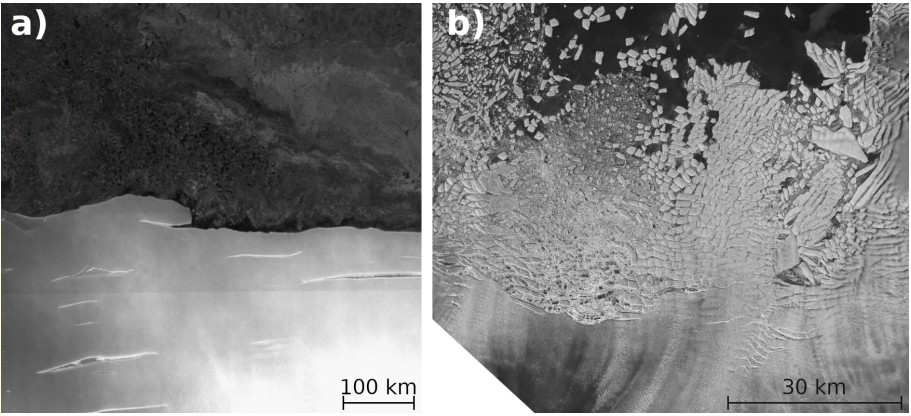

**Figure A3. a)** Part of the central Ross Ice Shelf front, photo from Copernicus Sentinel data 2023. Retrieved from ASF DAAC on 11/30/2023, processed by European Space Agency (ESA). **b)** Images of Thwaites Glacier Ice Tongue from April 2018 extracted from video by the Copernicus Sentinel-1 mission between 14 June 2017 and 7 July 2019, processed by ESA.

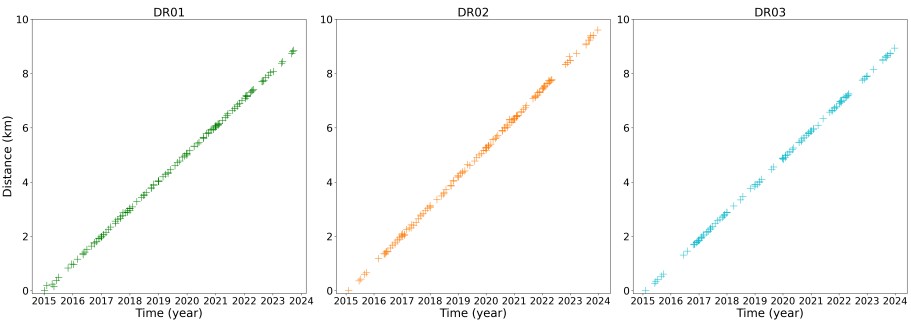

**Figure A4.** Front position for the free GPS buoy location from Sentinel-1 imagery data.

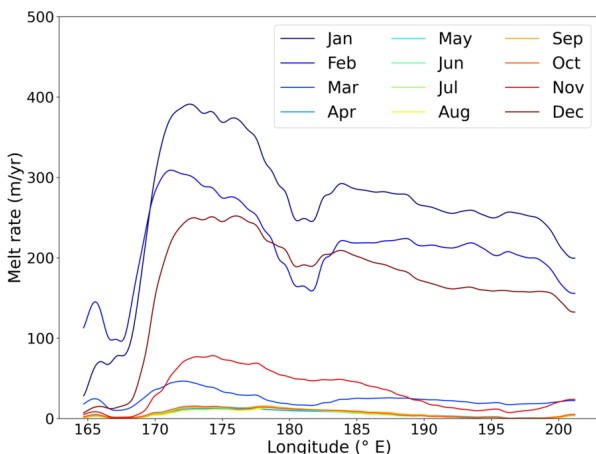

**Figure A5.** Wave induced melting along the Ross Ice Shelf front computed from the melt rates shown in Figure 3d. a) Climatological melt (averaged over 2003–2022) as a function of longitude, computed using the 60 km-wide near-front ocean swath of Figure 3. b) Zonally averaged climatology with the mean annual melt rate shown by the red line.

*Author contributions.* TJWW conceived the study idea. TJWW, NS, NP, and LKZ devised the methodology. NS carried out the analysis. MRS found and processed the ICESat time series of Figure 2. NS wrote the first draft with guidance from TJWW. All authors contributed to the interpretation of results and the writing of the manuscript.

*Competing interests.* The authors declare no competing interests.

*Acknowledgements.* The authors thank Maya Becker, Emily Glazer, and Roger Buck for helpful discussions. We acknowledge support from the NSF Office of Polar Programs through grants 2148544 and 2338057.

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
