# Peer review of "Wave erosion, frontal bending, and calving at Ross Ice Shelf"

_EGUsphere, 2024_

## Author Comment (AC1)

**Author Response to Reviewer Comments on: "Calving of Ross Ice Shelf from wave erosion and hydrostatic stresses" by Dr Ravindra Duddu (Reviewer #1) and an anonymous Reviewer (#2)**

**Authors in blue, Reviewers in black**

We would like to thank the reviewers for their insightful and constructive feedback. As there was significant overlap between the two reviews we have chosen to present our responses in the form of one combined document, however, we will address each individual comment separately below. We believe that the manuscript resulting from the revisions discussed below will be much improved from the original submission.

The primary weaknesses identified by the reviewers were (a) the absence of frontal pressure imbalances in the model and (b) the limited observational validation for wave-induced frontal melt and footloose calving. This prompted a deeper investigation into these aspects and revealed that the reviewers were spot on in their concerns. The additional insights have substantially improved our understanding of the system, and we believe strengthened the results of the study. Specifically, we addressed these concerns as follows:

- We incorporated a frontal bending moment term into the elastic beam equation. This term allows for the downward bending caused by the pressure imbalance at the front of ice shelves, as well as a bending moment resulting from the vertical viscosity gradient within the ice (following recent work by Buck (2024), see details below). The sum of these moments can be either positive (upward) or negative (downward), which leads to a marked improvement in the model's ability to match ICEsat-2 profiles, as seen in Figure R1 below. Moreover, this revised model allows us to parse the relative importance of both frontal bending moments and foot-induced shear forces. Notably, we identify frontal deformations that are most readily explained by a combination of both a bending moment and a foot-induced shear force at the front, a phenomenon that to our knowledge has not been previously observed.
- We integrated Sentinel-1 data (giving high-accuracy frontal position) with in-situ GPS station data from Klein et al (2020), (giving high-accuracy ice velocity) at several locations along the RIS front. This allowed for a much improved constrain on the total frontal ablation (resulting from the sum of frontal melting and small-scale calv-ing/frontal "crumbling"), as shown in Figure R2 below. To our knowledge, this presents the most accurate frontal ablation estimate available for RIS. We found that, as previously speculated, the wave-induced melt parameterization was overestimating the melt at the RIS front. We have added a discussion of how the melt parameterization can be improved for ice shelves.

In light of these changes we propose the revised title "Wave-erosion, frontal bending, and small-scale calving at Ross Ice Shelf".

**Reviewer 1 (Dr Ravindra Duddu):**

This article investigates the role of elastic bending stresses at the ice shelf front in driving

Figure R1: New Figure: Four ICESat-2 profiles with varying frontal deflections (black/gray markers). In each panel the observed profile is compared to the beam model without bending moment (red) and the model with a bending moment (green). The full model was fitted only to a subset of the observational data (black markers) to avoid issues from the increasing ice thickness with distance from the front. In panel (a) the inset shows a zoomed-in version of the frontal deflection, highlighting a combination of downward curvature and uplift that is most readily explained by a combination of a negative bending moment (from unbalanced ice-water pressure) and positive shear force (from a submerged foot). In each panel, the figure legend specifies the  $l_f$  and M parameters chosen to produce the best fit with the data. By design M = 0 for the simple model (red). Panel (d) highlights the case where two explanations give a close fit: one with a sizeable foot and no bending moment, and one with no foot but a positive bending moment (which could arise from internal stresses as suggested by Buck, 2024). We argue that disentangling these two processes may act as motivation for further investigations. The transect locations are (from a-d): (-78.25° N, -174.70° W), (-78.31° N, -171.79° W), (-77.39° N, 172.62° W), (-78.00° N, -160.11° W).

calving using the classical beam theory (Euler–Bernoulli equation) and compares it with satellite observations from ICESat and ICESat-2 elevation data. The novelty of this study lies in its focus on small scale calving arising from the footloose mechanism and parametrization of frontal wave erosion to estimate the size and frequency of calving mechanism. The article is well structured and easy to read, particularly, I liked how the discussion of model

---

## Author Comment (AC2)

**Author Response to Reviewer Comments on:**
**"Calving of Ross Ice Shelf from wave erosion and hydrostatic stresses"**
**by Dr Ravindra Duddu (Reviewer #1) and an anonymous Reviewer (#2)**

Authors in blue, Reviewers in black

We would like to thank the reviewers for their insightful and constructive feedback. As there was significant overlap between the two reviews we have chosen to present our responses in the form of one combined document, however, we will address each individual comment separately below. We believe that the manuscript resulting from the revisions discussed below will be much improved from the original submission.

The primary weaknesses identified by the reviewers were (a) the absence of frontal pressure imbalances in the model and (b) the limited observational validation for wave-induced frontal melt and footloose calving. This prompted a deeper investigation into these aspects and revealed that the reviewers were spot on in their concerns. The additional insights have substantially improved our understanding of the system, and we believe strengthened the results of the study. Specifically, we addressed these concerns as follows:

- We incorporated a frontal bending moment term into the elastic beam equation. This term allows for the downward bending caused by the pressure imbalance at the front of ice shelves, as well as a bending moment resulting from the vertical viscosity gradient within the ice (following recent work by Buck (2024), see details below). The sum of these moments can be either positive (upward) or negative (downward), which leads to a marked improvement in the model's ability to match ICEsat-2 profiles, as seen in Figure R1 below. Moreover, this revised model allows us to parse the relative importance of both frontal bending moments and foot-induced shear forces. Notably, we identify frontal deformations that are most readily explained by a combination of both a bending moment and a foot-induced shear force at the front, a phenomenon that to our knowledge has not been previously observed.

- We integrated Sentinel-1 data (giving high-accuracy frontal position) with in-situ GPS station data from Klein et al (2020), (giving high-accuracy ice velocity) at several locations along the RIS front. This allowed for a much improved constrain on the total frontal ablation (resulting from the sum of frontal melting and small-scale calving/frontal "crumbling"), as shown in Figure R2 below. To our knowledge, this presents the most accurate frontal ablation estimate available for RIS. We found that, as previously speculated, the wave-induced melt parameterization was overestimating the melt at the RIS front. We have added a discussion of how the melt parameterization can be improved for ice shelves.

In light of these changes we propose the revised title *"Wave-erosion, frontal bending, and small-scale calving at Ross Ice Shelf"*.

**Reviewer 1 (Dr Ravindra Duddu):**
This article investigates the role of elastic bending stresses at the ice shelf front in driving

[Figure]

Figure R1: New Figure: Four ICESat-2 profiles with varying frontal deflections (black/gray markers). In each panel the observed profile is compared to the beam model without bending moment (red) and the model with a bending moment (green). The full model was fitted only to a subset of the observational data (black markers) to avoid issues from the increasing ice thickness with distance from the front. In panel (a) the inset shows a zoomed-in version of the frontal deflection, highlighting a combination of downward curvature and uplift that is most readily explained by a combination of a negative bending moment (from unbalanced ice–water pressure) and positive shear force (from a submerged foot). In each panel, the figure legend specifies the $l_f$ and $M$ parameters chosen to produce the best fit with the data. By design $M = 0$ for the simple model (red). Panel (d) highlights the case where two explanations give a close fit: one with a sizeable foot and no bending moment, and one with no foot but a positive bending moment (which could arise from internal stresses as suggested by Buck, 2024). We argue that disentangling these two processes may act as motivation for further investigations. The transect locations are (from a–d): (-78.25° N, -174.70° W), (-78.31° N, -171.79° W), (-77.39° N, 172.62° W), (-78.00° N, -160.11° W).

calving using the classical beam theory (Euler–Bernoulli equation) and compares it with satellite observations from ICESat and ICESat-2 elevation data. The novelty of this study lies in its focus on small scale calving arising from the footloose mechanism and parametrization of frontal wave erosion to estimate the size and frequency of calving mechanism. The article is well structured and easy to read, particularly, I liked how the discussion of model

[Figure]

Figure R2: (New Figure) (a) Map of RIS near-front region. Indicated are the locations of GPS buoys DR01-03 from Klein et al (2020) in green, orange, and cyan, respectively. (b-d) Comparison of annual mean *ice flow velocity* (left) and *frontal advance velocity* (right) at the approximate locations of DR01-03. Shown is the ice velocity from the GPS buoy, located within $\mathcal{O}(1)$ km from the ice front (red) and the advance velocity of the front at that location as extracted from Sentinel-1 imagery (blue). The red error bars show interannual variability in the buoy data, the blue error bars show uncertainty in the frontal advance velocity, estimated using a parametric bootstrap method. The difference between the ice velocity and the frontal advance velocity gives an estimate of annual mean frontal ablation (black arrow). Also shown are frontal ice velocity estimates from MEaSURESv2 (purple). Note that these are underestimates of the ice velocity in (b) and (d), since physically the ice flow velocity can not be slower than the frontal advance velocity (this is presumably due to MEaSUREs relatively low resolution). Finally, in brown we show the frontal advance that results from the melt parameterization of Equation 6. This is computed by subtracting the estimated melt (Equation 6) from the buoy velocity. The brown vertical dashed lines illustrate the interannual variability in the melt estimate. The substantial discrepancy between the estimated advance (brown) and observed advance (blue) is a clear indicator that Equation 6 overestimates the melt. In panel (d) we also show a frontal advance velocity as given by the IceLines product (olive). Note that this must be overestimating the frontal advance since it is faster than the rather well-constrained ice velocity (red), which is unphysical.

limitations was weaved in quite nicely with the model parametrizations.

However, the main limitation of the article is that it uses the classical elastic beam theory equation and its analytical solution. The authors do acknowledge that viscous processes may play a crucial role and cite the work of Mosbeux et al. (2020). The article almost feels like a report on the limitations of simple model parametrizations, serving to bring awareness to the community that "Further investigations are therefore warranted . . ." as remarked in the

[Figure]

Figure R3: Distribution of buoyancy wavelength ($l_w$), foot length ($l_f$), and the maximum tensile stress ($\sigma_m ax$) for the two models formulations. The model with nonzero frontal bending moment is depicted in green for negative moments and in blue for positive moments, with the two quantities stacked. The foot-only model is represented in red.

conclusion. While the article does not advance the state-of-the-art models or propose new parametrizations, I do think it would make a contribution to the literature. My detailed comments are listed below.

We thank the reviewer for this feedback. As discussed in this document, we have aimed to expand the scope of this study to go beyond reporting on the limitations of previous parameterizations/simple models, which was certainly a weakness of the original submission.

**1/** Line 25 – Two review papers (Benn et al., 2017, Alley et al., 2023) were cited. I think a recent article Bassis et al. (2024) discussing the stability of ice shelves and ice cliffs in changing climate would be a relevant reference. `https://www.annualreviews.org/content/journals/10.1146/annurev-earth-040522-122817`.

We thank the reviewer for this citation recommendation. We have added this reference, as it is highly relevant.

**2/** In equation (1), why was the bending moment and axial compressive load at the ice shelf front due to the (linearly-varying) hydrostatic water pressure not considered. Instead, only a point load arising from increased buoyancy of the foot is considered. Is the moment neglected because there is no analytical solution like equation (2) for the case when moment is included. On line 137, it is remarked that the moment is neglected for simplicity and cited Mosbeux et al. (2020) that this will cause 15 – 20%. It would be great to see some more details and a quantitative analysis of the effect of the end moment and compressive pressure at the ice-ocean front in this paper, especially given the compressive pressure often tends to

[Figure]

Figure R4: Comparison between the wave-induced melt using the revised sea-ice expression in the melt parameterization (black) and the original parameterization (gray). The standard deviation of temporal variability is represented by a faint blue envelope around the revised parameterization, with the annual means indicated by horizontal lines.

stabilize ice cliffs.

Thank you for highlighting this key point. As mentioned above, this comment rightly identifies one of the primary weaknesses of the original model, which we have addressed by incorporating the bending moment into the model (see equation below with moment $M$). We had not appreciated the importance of this effect in this context. For icebergs (which we studied previously with this framework), the frontal bending moment appears to be less impactful. Crucially, incorporating the moment allows a more accurate match with the RIS front profiles from ICESat-2 (cf. Figure R1), suggesting that bending moments indeed play a significant role in certain cases. While including this effect had a notable impact on explaining the frontal deformations, the impact on the conclusions regarding calving is less pronounced: calving will mostly occur when bending stresses are so high that you need a rather large foot. In that limit, the effect of the additional moment on calving lengths and frequencies is relatively small.

The solution of the beam subject to a foot and a bending moment can then be written as

$$w(x) = l_w e^{-\frac{x}{\sqrt{2}l_w}} \left[ \left( M + \sqrt{2}\mathcal{H}\frac{l_f}{l_w} \right) \cos\left( \frac{x}{\sqrt{2}l_w} \right) - M \sin\left( \frac{x}{\sqrt{2}l_w} \right) \right], \tag{1}$$

where $M = l_w \frac{d^2 w}{dx^2}\big|_{x=0}$. If the frontal bending moment only arose from the difference in ice burden and water pressure at the front, $M$ could be estimated from the geometric conditions (as is done, for example, in Wagner et al, 2016, and Slater et al, 2021). However, it was recently shown by Buck (2024) that the nature of this bending moment is more complicated and can be positive when the vertical temperature gradient in the ice is large enough. This results in a complex set of drivers for the bending moment, with large uncertainties. We realized it may be more helpful to treat both $M$ and $l_f$ as unknown parameters that can be

adjusted to match the observed ice shelf profiles. This results in close agreement for a large variety of profiles, including profiles that feature berm-like features at the front. Overall, the results become considerably more insightful than with the simple foot model. Four examples of observed profiles and model solutions (with and without bending moment) are shown in Figure R1 below, which will be added to the manuscript.

We have recalculated the estimated values for foot length, $l_f$, buoyancy length, $l_w$, and resulting maximum stress, $\sigma_{max}$, shown as histograms in Figure R3. Estimates in buoyancy lengths are largely unchanged, as is the distribution of finite-sized $l_f$ and associated estimates of maximum stress.

The main change in the results is that we now find that $\sim 170$ profiles (of 928 total) are best explained by a pure upward bending moment at the front (akin to the profiles in Buck, 2024) and $\sim 20$ profiles are best explained by a pure downward bending moment as one would get in the classic berm shape (neither of these were accommodated in the simple foot model). These pure bending moment profiles have predominantly small to modest frontal uplift ($< 5$ m) and the location of highest stress is at the front, $x = 0$. As a result, these are unlikely to contribute to calving. For $l_f \gtrsim 10$ m (the regime most relevant to the calving question) the revised model gives results that are well approximated by the simple foot model, with the deformation largely dominated by the foot and small positive or negative bending moments playing a secondary role.

We note that, while the model remains analytically solvable with the bending moment, this addition complicates some of the equations. Since the models converge in the limit of large feet, which is the limit most relevant to calving, we have retained the estimation of calving sizes using the foot model, since this allows for the most intuitive presentation of the discussion.

Regarding the tension resulting from compressive pressure, we tested its impact on the model by numerically solving a model that includes both bending moment and compressive tension. This analysis showed minimal impact on shelf deflection. Therefore, to maintain the model's analytical solvability, we decided to omit the effect of tension (in line with similar treatments by Slater et al 2021, Buck 2024).

**3/** Going from eq. (1) to eq. (2), please provide the expression used for the foot-related forcing term Q. It is not clear how equation (3) was obtained. Also, below equation (4) it is stated that $x_{RM}$ does not depend on the size of the foot. Perhaps, these two things were clarified in Wagner et al. (2016), but would be good to add a clarification to this paper.

Thank you, we agree that these clarifications will be helpful and have added the expression of $Q = (\rho_w - \rho_i)gl_f d$ in the text to lay out how equations (3) and (4) are obtained. We also now detail how equation (4) was derived in the appendix. We solve it for the main case, which includes a general frontal moment $M$, and derive the particular case when the moment is small which gives equations (3) and (4).

**4/** Line 108 – Poisson's ratio of 0.3 is used throughout this work, modelling ice as being compressible. Wouldn't a Poisson ratio of 0.5 be more appropriate for the time-scales considered (years), as this would result in the ice being incompressible.

Thank you for this comment. Since we're considering a purely elastic framework here, we have retained a Poisson ratio of 0.3. We note that previous studies by Christmann et al (Annals of Glaciology, 2016) and Mosbeux et al (The Cryosphere, 2020) have found that changing to a larger ratio of 0.4 or 0.5 tends to have a small effect of $< 5\%$ on the magnitude of maximum tensile surface stress. A comment noting this will be added to the manuscript.

**5/** Line 132 – Following Wagner et al. (2014) a simple yield stress based criterion is implemented for the calving event. I am afraid this is overly simplistic. In a recent article (Gao et al., 2023), we have proposed an advanced finite element based cohesive zone models for simulating nonlinear viscous and hydrofracture process in ice. I suggest the authors could perhaps acknowledge the need for some advanced process models for better representation and understanding of near-terminus small-scale calving events. `https://ieeexplore.ieee.org/abstract/document/10271321`

We appreciate that we didn't sufficiently caveat the use of such a simple criterion. While we retained this criterion to maintain the analytical approach, we're adding a statement to the revised manuscript to recognize this limitation and motivate its use within the idealized framework of the study. We're now referring to the above mentioned paper to account for the complexity behind the fracture process.

**6/** Section 3.2 both describes the process creating the foot as "erosion" and "melting". Please clarify whether this is driven by melting (thermal energy being advected to the ice-shelf to induce a phase-change) or erosion (wave motion mechanically removing material through impacts/abrasion).

We agree that this ambiguity requires clarification and we thank the reviewer for raising this issue. We are revising the sentence to clarify that "wave erosion" here refers to the thermal melting of the notch, in keeping with the original coining of the term "wave erosion" in this context by White et al., "Theoretical estimates of the various mechanisms involved in iceberg deterioration in the open ocean environment" *(USCG-D-62-80 Final Rpt, National Technical Information Service, 1980)*, and common usage in the literature since (e.g., Silva et al., 2006; Stern et al., 2016). The concept of mechanical abrasion has not been discussed in this context and we eschew the associated complexities here.

**7/** Line 162 – It is stated that equation (6) has not been validated comprehensively against real world conditions. Please clarify whether this is in the context of calved icebergs as

done by Gladstone et al. (2001) with the parameters from Martin and Adcroft (2010) and England et al. (2020), or is their statement more about ice shelves.
I think it is a missed opportunity that the authors have not made any advancement on this, but rather applied an empirical formula from the literature. At the least a discussion on how to improve the state-of-the-art in the context of wave erosion/melting would be useful.
Also, is the use of the "mean over an ocean strip along Ross Ice Shelf" well justified by the following statement that melt rate estimates are not sensitive to the choice of the ocean strip width. I found these sentences a bit confusing, so please rephrase and clarify.

Thank you. The comment raises three points:

- We agree that clarification is needed regarding the origin of Equation 6. We have revised the sentence to clearly state that Equation 6 was developed for icebergs and has not been previously applied to ice shelves. Wave-induced melting of ice cliffs remains an active area of research, and after conducting a thorough literature review, we concluded that this equation is currently the best available, given the similarities in physical processes between icebergs and ice shelf fronts in this context.

- We agree that not evaluating the melt parameterization would have been a missed opportunity. As outlined in our primary response, we utilized satellite and GPS station data to constrain frontal melting better (Figure R2). Additionally, we realized that the melt equation's dependence on sea ice concentration as originally put forward by Gladstone et al (2001) is likely unsupported. We found that a linear dependence on sea ice concentration produces a better fit with observed wave attenuation in sea ice. We therefore replaced the cubic exponent in the cosine term of Equation 6. This adjustment helps improve the melt rate values, particularly during winter where minimal melting is expected (seen in Figure R4). Finally, we conclude that the melt parameterization as used in the literature overestimates the actual melting by a factor of 5-6, and argue for further work to better constrain this.

- Thank you for highlighting the lack of clarity in this paragraph. We have revised it to better lay out that we compute the mean over a given ocean strip since the wave melt-rate parameterization is local. In order to avoid small errors associated with "bleeding" effects from the coastal grid boxes we average over a finite strip width. We confirmed that the exact choice of the strip width has minimal impact on the results.

8/ Line 170 – two dataset resolutions are specified in degrees and the other one for sea ice concentration is specified as 25 km. Please clarify what a 0.01 degree resolution means in terms of 25 km resolution, and various quantities were can regridded without interpolation.

Thank you for this valuable suggestion – we have updated the relevant paragraph to include both the grid resolution in degrees and the corresponding distances in kilometers. Regarding the choice of regridding without interpolation, we opted to avoid artificially smoothing the data, given the relatively different grid sizes within the dataset. This decision results in sharper transitions between the grid cells of the largest data grid (25 km), which remains

small in comparison to the overall size of the Ross Ice Shelf (RIS) front.

**9/** Line 204 – 207 – Several of the profiles having berm characteristics were excluded in Figure 4a showing Ross Ice Shelf elevation profiles. Please add a sentence on why these profiles exist and why the rampart-moat profile is not found everywhere. Does this correspond to locations where a small scale calving event occurred in the past leaving behind the berm shaped profile. Is it possible to look for the elevation data for berm profiles across time. Please clarify how much of the data-set was excluded by the criteria (presumably this is only a small portion of the dataset?) If not, please comment on how representative are the presented results if a significant portion of the reference data does not follow this mechanism?

The reviewer is correct in this interpretation and we appreciate the request for clarification here. We have added the following sentence to explain why the rampart moat is not consistently found along the RIS front: "As shown in Figure A1, berm profiles (in black) or small ramparts (in yellow) are typically observed in patches along the RIS front. This pattern may suggest that local factors, such as high basal melt, prevent the formation of the foot, or that recent calving events have locally removed any trace of it."

Regarding the time series of elevation profiles, the only identified profile that exhibits an unequivocal calving sequence is the ICESat-1 series shown in Figure 1. Additional profile series may be identifiable within the ICESat-2 data, but the task is more challenging than may be expected, due to a number of factors including the large data volume paired with sporadic coverage, small vertical signals at the early stages of the cycle, and stochastic nature of calving events. In particular, given that berm profiles constitute a small proportion of the dataset, challenges may arise when attempting to find multiple transects taken repeatedly at the same location and featuring berms.

Lastly, rampart–moat profiles with a rampart height smaller than 2 meters represent 30% of the data, which is not insignificant. Although these profiles align with the model presented for the rampart section, they were excluded due to the noise introduced by the small uplift when scaled by small $w_{RM}$. Following the inclusion of the bending moment in our model, we have remade Figure 4 to include all profiles that feature a rampart-moat expression. The berm-dominated profiles were still excluded from Figure 4 as they cannot be scaled similarly to the rampart-moat profiles. An example profile with a (small but clear) downward frontal bend is shown in Figure R1. The remaining berm profiles are included in the appendix.

**10/** Just a comment - Banwell et al. (2019) assumed 1 GPa as well, which gave better match of vertical deflection of ice shelf due to flexure from surficial lakes. As remarked by the authors, I agree that the discrepancy between laboratory and field/observational values may be arising from viscous creep effects https://www.nature.com/articles/s41467-019-08522-5

Thank you for your insightful remarks and for directing our attention to the relevant paper. A reference to this work has been added.

**11/** Line 224 – It is good that effect of firn layer is accounted by averaging the ice density to 850 $kg/m^3$. In Gao et al. (2023) our major finding was that deeper crevasses are possible in ice shelves because the firn layer increase the height of ice above the sea level. Such crevasses will of course change the effective ice thickness and the apparent flexural rigidity of the ice shelf, as remarked by the authors in lines 241 – 242. `https://ieeexplore.ieee.org/abstract/document/10271321` Also, Line 245, I would think crevasse spacing also effect the flexural rigidity not just crevasse depth.

Thank you for this insightful comment. We included the density of firn in the model because it seems to slightly improve the overall results. Additionally, in response to your suggestion, we have now incorporated a discussion of crevasse spacing as a factor that could reduce flexural rigidity.

**12/** Line 252 – Just curious if there is any physical interpretation for the quadratic scaling proposed for effective ice thickness h*

The motivation behind this scaling was that generally a power-law relationship is physically plausible, as we may expect crevasses to have a greater impact on thinner ice shelves. The exponent 2 then produced the best fit. However, following the changes to our results from the revised model, we have removed the quadratic scaling discussion from the manuscript.

**13/** I found Section 4.2 a little bit difficult to follow. Specifically, I did not follow how the authors determined L $\sim$ 110 m, $l_f = 0 - 32$, and also how they judged $l_f^{max} = 30$ is consistent with the image of the calved iceberg in Fig. 1c. Also, from the histogram in Figure 6, the maximum foot size is about 30 m and is smaller than the theoretical critical foot size of $l_f^{max} = 44$ m for 50 kPa. The authors state "bending stress alone may not be sufficient to initiate crevasses" Are the authors referring to perhaps water pressure in basal crevasses or just nonlinear viscous deformation induced stresses. Please clarify.

We want to thank the reviewer for pointing out the unclear paragraph. Following the new insights, we restructure Section 4 and partially rewrite Section 4.2 to improve its clarity; the updated version will be arranged as follows:

- 4.1 - Observations of Frontal Ablation: Constrain the total ablation rate at the RIS front by differencing frontal positions (from Sentinel-1) and comparing them to ice velocity (from GPS buoys), illustrated in a new figure (Figure R2).

- 4.2 - Beam Theory Fit to Observations: Demonstrate that the beam model captures the observed ICESat-2 profiles, now incorporating both submerged feet and frontal bending moments. This will be illustrated with the new Figure R1 .

- 4.3 - Calving Length: This section will now focus solely on the predicted calving lengths

from the two models (with and without the frontal bending moments). Here, we show that the estimates of $l_w$ and $l_f$ do not substantially differ between the two model formulations. In particular for the case of larger feet, the maximum stresses $\sigma_{max}$ of the two models are comparable (see Figure R3). This allows us to proceed with the simple foot model to estimate calving rates.

- 4.4 - Calving Frequency:

  - 4.4a - Maximum Foot Length: Drawing on the relationship $l_f(\sigma_{max})$ derived in Section 4.3 and applying the (appropriately caveated) calving criterion $\sigma_{max} \geq \sigma_y$ allows us to establish an estimate for $l_f^{max}$.
  - 4.4b - Foot Growth Rate: Here, we analyze the wave-induced melt rate and compare the results with those from Section 4.1. We discuss what realistic melt rates at RIS are (from Section 4.1.), how the original parameterization overestimates melt rates, and how a new parameterization should be constrained.
  - 4.4c - Estimated Calving Frequency: Estimate the calving frequency based on the above analyses, now combining the estimate for $l_f^{max}$ with the estimated melt rate from the observation (rather than the parameterization).

- 4.5 - Calving Rate: Integrate the preceding constraints on calving lengths and frequencies to estimate a foot-induced calving rate.

Regarding the comparison with Figure 1c, it is challenging to accurately determine the size of the iceberg's foot from the photograph due to the absence of a reference point and the iceberg's plan not parallel to the background plane. However, by using the flag on the left as a reference, we estimate the foot's size to be approximately 20 to 80 meters.

Finally, as rightfully pointed out by the reviewer, we clarify the sentence on the initiation of crevasses through nonlinear viscous deformation.

**14/** In Section 4.4, the authors state ". . . merely intended as back-of-envelope estimates." Please add a statement to clarify why these estimates are still useful, perhaps from the point of view of global ice sheet model parametrizations. I also commend the authors for their honest statement that the agreement between the theoretical and observed calving rates is somewhat of a coincidence. See my comment below on the calculation of the average calving rate. Although it is good to acknowledge viscous process, the Deborah number calculation is defined for linear viscous material, whereas ice is nonlinear viscous with strain rate dependence. This means that the timescale of relaxation (or bending) can be heterogeneous depending on the local strain rate in an ice shelf or glacier. We address this in a paper under review in The Cryosphere `https://egusphere.copernicus.org/preprints/2024/egusphere-2024-346/`

We are grateful for the reviewer's pertinent comment. In response, we have modified the quoted sentence in Section 4.4 to clarify that one aim of this study is to estimate the order

of magnitude of the footloose calving's impact on the RIS mass balance, partly because we deem it interesting of itself and partly to aid determining whether it should be included in calving parameterizations of ice shelf modes. Additionally, we appreciate the remarks regarding the Deborah number calculation, and we have added a comment to acknowledge the issue of heterogeneous relaxation timescales, citing the relevant reference provided by the reviewer.

**15/** A simple fluid mechanics (mass balance) compatible calving rate law is implemented here. This calving rate law is typically defined by averaging over many calving events, thus more suited for glaciers or ice shelves that regularly calve ice bergs (see discussion in Section 6.1 in Bassis et al., 2024). However, in the case of the Ross ice shelf one calving event occurred between 2006 and 2007, which was averaged over span of 4 or 5 years to obtain the average calving rate of 200 – 300 m/yr. I find this a bit odd and almost seems like the model was forced to fit to the data. In my opinion, a calving rate law is not well suited to describe the footloose calving mechanism, however, large scale ice sheet models seem to use this simplistic calving laws. It would be great if the authors can add some more discussion on this in the paper. https://www.annualreviews.org/content/journals/10.1146/annurev-earth-040522-122817.

We appreciate the reviewers helpful input on this complex question. The observed calving event in ICESAT-1 data of size ∼1km is likely the result of a specific combination of foot-induced stresses and crevasse location, and we agree that extending this to a continuous average calving rate is not overly helpful. The reference to this average calving rate has therefore been removed and we clarify the likely uniqueness of this time series. In the revised manuscript we expand on the more robust frontal ablation rate of ∼20 m/yr and discuss the potential of this being due to a combination of wave-induced melting and smaller-scale calving events—some of which we argue will be a result by foot-induced stresses.

**16/** Lines 328 and 329, I did not understand how theoretical and observation process times $\tau$ were calculated as 0.5 and 5 years. Also, if theoretical estimate is 0.5 then should the Deborah number range from 0.3 – 40 instead of 0.3 – 10.

Thank you for pointing out this lack of clarity and typo in the $De$ estimate. We have revised this paragraph to lay out the derivation of the $De$ range, which, following the revisions to the model and observational constraints is now $De = 0.03 - 5$.

**17/** Line 337 – the phrase "... maximum location of maximum stress ..." is confusing. Can this be rephrased as – the location of global maximum of principal stress? By maximum stress are the authors referring to the maximum principal component and its global maximum?

We thank the reviewer for pointing at this confusing sentence. We have clarified it to read "the location of global maximum tensile stress".

**Reviewer 2:**
This paper makes the innovative proposal that the 'footloose' calving mechanism – well-established in Arctic contexts – might be an important process of mass loss from the Ross Ice Shelf. The paper is well structured and easy to follow, and basically proceeds in three stages of unequal length: 1. presentation of observational evidence for the 'footloose' process; 2. a simple model of the process is presented then used to predict calving rates, and 3. discussion of model performance and validation. Some of these are much more convincing than others.

**1/ Evidence for the process:** Evidence for the process is threefold: 1. A photograph of an iceberg provides striking evidence for the existence of an ice foot. 2. Good use is made of ICESat elevation data to show that rampart and moat profiles are common, providing convincing evidence of 'bottom-out' flexure of the ice front. 3. One example is given of a single calving event in a six-year ICESat record, which appears to have cut off the ice shelf at the point of maximum depression. It is not explicitly stated that this is the only example found in the record, but it seems reasonable to assume that it is. The evidence thus provides support for the idea that ice feet form at the margin of the Ross Ice Shelf, and that these cause buoyant flexure. The evidence also supports the idea that 'footloose' calving does remove ice from the Ross Ice Shelf, although the fact that this evidence is confined to a single set of ICESat profiles and one archive photograph indicates that such calving events are rather infrequent.

We thank the reviewer for this summary and for highlighting a couple of weaknesses in our initial presentation: First, yes, the shown ICESat-1 timeseries featuring the calving event is indeed the only sequence of tracks that shows an almost complete calving cycle. Second, we agree that these calving events are likely rather infrequent, which was lent additional support by the new constraint of ∼20 m/yr frontal ablation. This entails that, while the wave erosion and foot-induced frontal bending at the RIS appears to be a complex, striking, and wide-spread phenomenon, it likely only accounts for 2-3% of the total RIS mass loss. This supports the idea that the dominant mass-loss process is the sporadic rift-driven calving of giant icebergs. We have adjusted our discussion and conclusions to reflect these findings. We further argue that, even though this process may not be a major driver of RIS mass loss, investigating the dynamics of wave erosion and hydrostatic deflections of ice shelves is an important step toward better understanding calving dynamics both at ice shelves and tidewater glaciers (see e.g., Slater et al, 2021; Buck, 2024). Buck (2024) further makes the interesting point that these deflection phenomena may shed light on ice rheology questions.

**2/ The model:** This is the least satisfactory part of the paper. As it stands, the model is highly simplified, and has the major omission that it neglects the effect of depth-varying back-pressure on the ice front. Uneven back-pressure is well known to cause 'top out' flexure of floating ice fronts, and thus opposes the 'bottom out' flexure due to the development of an ice foot. Since the two flexure processes oppose each other, it is essential to establish their relative importance as an ice foot grows, as this will determine the stress distribution in the

ice and ultimately, the critical ice foot length required for calving. Because it neglects this process, the model is of little value in either an illustrative or predictive sense.

We want to thank the reviewer for raising this weakness in the study methodology. As also mentioned in response to Reviewer #1, our neglect of the frontal bending moments was an artifact of having applied this framework previously to icebergs. It appears that for icebergs, particularly those in open and warm waters, feet can grow rather quickly to relatively long lengths and the resulting uplift dominates over the frontal bending moment. Following the two reviewers' suggestions we included this effect in the model and have found that it allows us to much better capture the wide range of ICESat-2 altimetry profiles. The new results provide interesting insights into the relative roles of bending moments and foot-induced deflection. In line with the recent work by Buck (2024) it appears that internal bending moments may often act in tandem with underwater feet. The revised model is able to partition between the different effects, as shown here in Figure R1. We refer the reviewer to our response to Comment 2 by Reviewer #1 for a detailed discussion of the revised model results.

To model the footloose process properly would probably require the use of a finite-element model such as Elmer/Ice to explore the evolution of elastic stresses and viscous deformation during the growth of an ice foot. Such an approach would have the advantage of identifying the actual locations and magnitudes of stress concentrations, which would provide a much more secure foundation for calving predictions.

We thank the reviewer for pointing this out, and we readily acknowledge that the model used is very simplistic given the complexity of the physical processes involved in ice shelf deformation and calving. One of us (TJWW) has previously worked on the comparison between viscous and elastic effects in footloose-type deflections, published in Mosbeux et al (2020). For the present study, the primary objective was not so much to provide realistic simulations of the calving process, but rather to get a broad assessment of how frontal wave-erosion may drive hydrostatic stress imbalances which then leads to deflections and potentially calving. Our aim was to identify some of potentially driving processes that would explain the observations discussed above, and we would argue that to this end an idealized model can be more insightful than a more comprehensive approach, which can be somewhat "black-boxy". Having said this, the omission of a frontal bending moment was evidently one simplification too far, and we are grateful for the reviewer to point that out, as it has allowed for a much improved interpretation of the observations.

**3/ Validation:** In Section 4.4, there is some discussion about the differences between the results and the (limited) available data for validation. The text notes that there are 'some differences' but the differences are really quite major: calving frequency is evidently much less than the 2/yr suggested by the model, and the single example of calving in the ICESat data suggests that calving magnitudes may be typically much greater. Furthermore, the brief discussion of ice-front advection vs. ice velocity indicates that the calculated frontal wave erosion rate is too high. Taken together, these results indicate that the model does a poor

job of describing the process and predicting calving rates. There is thus little justification for the claim that the footloose mechanism accounts for "up to 25%" of frontal ablation of the shelf.

We thank the reviewer for raising these important points which also highlighted some more general issues with the clarity of presentation of Section 4. In response, we have reorganized this section to improve clarity and to lay out more clearly which observations were used to (a) test the melt parameterization, (b) constrain the beam model, and (c) assess the calving frequencies and volumes associated with this process. (Please see our response to Comment 13 from Reviewer #1 for further details of how we have restructured Section 4).

We have aimed to address the weaknesses pointed out by the reviewer where we could and justify instances where expanding the observational analysis or increasing the model complexity would go beyond the scope of this study (e.g., adding a visco-elastic account, which will certainly be a valuable follow-on contribution). Particularly, the original calving frequency of 2 per year was primarily influenced by the high melt rate derived from Equation 6. Our new constraints on total frontal ablation (from Sentinel-1 and GPS buoys, as discussed above) suggest that this melt rate is an overestimate, and more realistic values would result in more accurate calving frequencies when applying the elastic beam model. As noted above, we have constrained the wave-induced melt rate and refined its formulation, leading to more consistent results. Additionally, we have updated the manuscript's conclusion to reflect these improvements.

**4/ Conclusion:** Having said this, the idea is an interesting one and I think the paper should be published in some form. I strongly suggest, however, that the authors make a big effort to improve the model. This will provide a more convincing picture of how the footloose process might actually work in an Antarctic context and will help identify the key factors that control calving losses. I recommend either one of two possible ways forward. Publication could be delayed until more work is done, especially with regard to developing a better model of the footloose process. Alternatively, the paper could be shorn of much of the modelling element to focus on the observational evidence that ice-foot development, bottom-out flexure, and footloose calving actually operates on the Ross Ice Shelf. In both cases, quantitative data on ice velocities and frontal position should be compiled to quantify cliff retreat rates. Whichever course is taken, I think that major revisions are required before publication.

We appreciate the thorough and helpful review and comments on the manuscript. We believe that incorporating the frontal bending moment due to in-plane stresses—as suggested by the reviewer—presents a substantial improvement that allows us to parse the roles of frontal bending and shear forces, while still providing a representation that is simple enough to shed fundamental insights. Furthermore, the initially predicted melt rate was significantly overestimated. To address this, we combined satellite and GPS station data to constrain the wave-induced melt rate and modified the melt parameterization to improve winter estimates. These improvements have significantly enhanced the overall quality of the model and achieved better consistency with the observations.